# Calving Front Machine (CALFIN): Glacial Termini Dataset and Automated Deep Learning Extraction Method for Greenland, 1972-2019

**Daniel Cheng**[1], **Wayne Hayes**[1], **Eric Larour**[2], **Yara Mohajerani**[1,3], **Michael Wood**[2], **Isabella Velicogna**[1,2], **and Eric Rignot**[1,2]

[1]University of California at Irvine, Irvine CA, USA
[2]Jet Propulsion Laboratory, California Institute of Technology, Pasadena CA, USA
[3]University of Washington, eScience Institute and Department of Civil and Environmental Engineering, Seattle, WA, 98195, USA

**Correspondence:** Daniel Cheng (dlcheng@uci.edu)

**Abstract.** Sea level contributions from the Greenland Ice Sheet are influenced by the rapid changes in glacial terminus positions. The documentation of these evolving calving front positions, for which satellite imagery forms the basis, is therefore important. However, the manual delineation of these calving fronts is time consuming, which limits the availability of this data across a wide spatial and temporal range. Automated methods face challenges that include the handling of clouds, illumination differences, sea ice mélange, and Landsat-7 Scanline Corrector Errors. To address these needs, we develop the Calving Front Machine (CALFIN), an automated method for extracting calving fronts from satellite images of marine-terminating glaciers, using neural networks. The results are often indistinguishable from manually-curated fronts, deviating by on average 86.76 ± 1.43 meters from the measured front. Landsat imagery from 1972 to 2019 is used to generate 22,678 calving front lines across 66 Greenlandic glaciers. This improves on the state of the art in terms of the spatio-temporal coverage and accuracy of its outputs, and is validated through a comprehensive intercomparison with existing studies. The current implementation offers a new opportunity to explore sub-seasonal and regional trends on the extent of Greenland's margins, and supplies new constraints for simulations of the evolution of the mass balance of the Greenland Ice Sheet and its contributions to future sea level rise.

## 1 Introduction

The evolution of Greenland's tidewater glaciers is an important constraint on the evolution of the Greenland Ice Sheet (Nick et al., 2013). Likewise, changes in Greenland are important in tracking and predicting future sea level rise over the next century (Andersen et al., 2015; Fürst et al., 2015; van den Broeke et al., 2016). Constraining Greenland's glacial evolution is thus an important part of improving the understanding of the earth system as a whole. One constraint on glacial evolution is the position of glacial calving fronts and ice margins over time (King et al., 2018). While satellite imagery allows for the extensive documentation of this evolving constraint, most calving front delineation is still done with time-consuming manual labor (Carr et al., 2017; Bunce et al., 2018; Catania et al., 2018). This results in the under-utilization of available satellite imagery, and causes gaps in seasonal records that introduce uncertainty when modeling past and projected climate change (Catania et al., 2020). Significant efforts have been made to improve this situation, which include the ESA-CCI dataset of 26 Greenlandic glaciers from 1990-2016, the PROMICE dataset of 47 glaciers from 1990-2018, and the MEaSUREs dataset of 200+ glaciers from 2000-2017 (ENVEO, 2017; Andersen et al., 2019; Joughin et al., 2015). Yet the increasing availability of new datasets through missions like Landsat 8 and the release of old datasets through improved reprocessing call for new automated ways of detecting the calving front. In particular there is a strong need for these automated ways to be robust, specifically against cloud cover,

ice mélange, shadows, and Landsat 7 Scanline Corrector Errors. Traditional automated techniques such as the edge detection utilized by Seale et al. (2011) and Paravolidakis et al. (2016) have significant challenges with respect to these issues. Modern machine learning techniques and deep neural networks provide a robust, scalable, and accurate solution to these processing challenges. Existing work by Mohajerani et al. (2019) pioneers the usage of these techniques by applying the Ronneberger et al. (2015) UNet deep neural network for Jakobshavn, Helheim, Sverdrup, and Kangerlussuaq Glaciers. It achieves a mean distance error of 96.3 m, but is restricted by the preprocessing requirement of aligning the flow direction to be vertical, and inability to handle branching/non-linear calving fronts. Zhang et al. (2019) evaluates a modified UNet applied to TerraSAR-X data over Jakobshavn Glacier, and achieves a mean distance error of 104 m, but is limited in scope. Baumhoer et al. (2019) expands the application of the UNet to Sentinel 1 imagery of Antarctica, extracting full coastline delineations and achieving a mean distance error of 108 m. Ultimately, these case studies provide the groundwork for the automatic, accurate, large scale, long time-series, high temporal resolution, and potentially multi-sensor extraction of glacial terminus positions. This study seeks to assess the feasibility of achieving robust automatic extraction for a selection of Greenland's glaciers, and to provide the resulting dataset for use by the wider community. Additionally, this study seeks to assess improvements to the neural network design and post-processing methods.

In this study, Sect. 2 covers the data source along with the spatial and temporal coverage. Sect. 3 examines the CALFIN algorithm and method for processing the data. Sect. 4 validates the algorithm through error analysis. Sect. 5 and Sect. 6 show and discuss the results - the calving front dataset and algorithm.

## 2  Data Source and Scope

For the production of the CALFIN dataset, Landsat optical images are used for their long time-series availability and reasonable spatial distribution/resolution. The area of interest for the dataset production is restricted to Greenland, in particular the calving fronts for 66 Greenlandic basins shown in Fig. 1, spanning the 1972 to 2019 time period shown in Fig. 2. The basins are selected for their high discharge volumes, wide spatial distribution, and diverse morphological features. The product used is the 60/30 meter resolution Near Infrared band. The 15 meter resolution panchromatic band was not used, due to computational and logistical limitations. A unique characteristic of this data source is the presence of Landsat 7 Scanline Corrector Errors from 2003-2013, which manifests as black stripes that interfere with automated calving front extraction methods.

For the training and validation of the CALFIN methodology, TerraSAR-X and Sentinel 1A/B SAR images are added to enforce the applicability of the method across different sensors and domains. The area of interest for the training and validation of the methodology thus includes Antarctic SAR data in addition to the Greenlandic Landsat optical data (see Sect. 3.2 and Fig. S4). The TerraSAR-X product used is the StripMap 3 meter resolution HH polarization band. The Sentinel 1A/B product used is the Extra Wide Swath, Ground Range Multi-Look Detected, 40 meter resolution HH polarization band. The other data products and polarization bands are not used since the backscatter intensity provides sufficient information for the data processing methodology to succeed. A characteristic of SAR data is the presence of speckle noise, which is addressed by the methodology described in the following section.

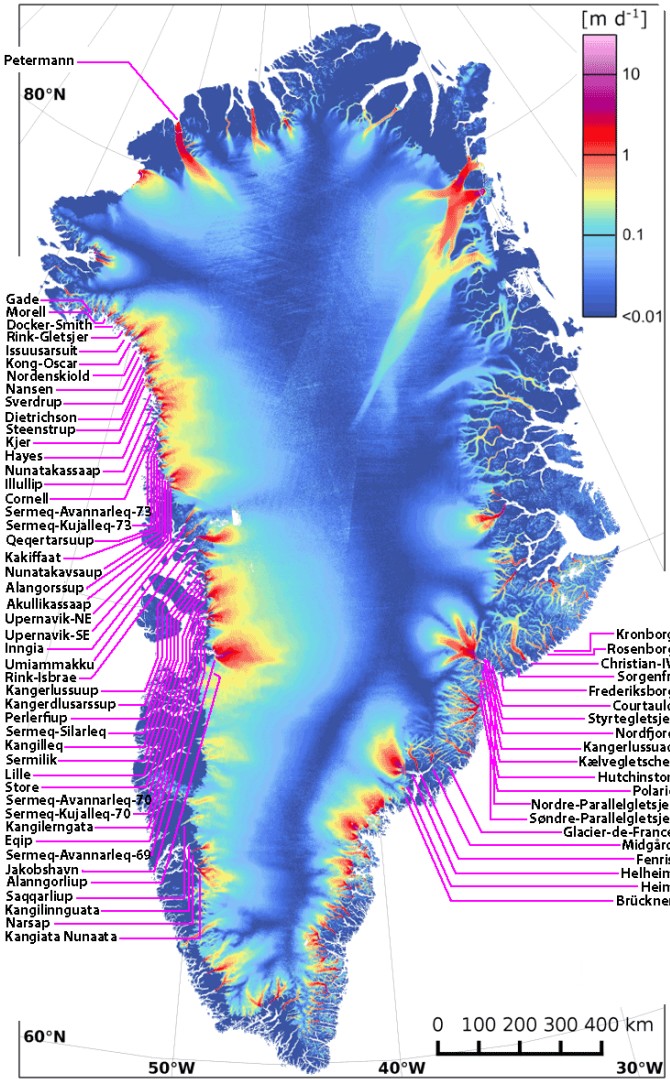

**Figure 1. Spatial Coverage Map**: Spatial distribution of 66 selected Greenlandic glaciers. The velocity map is taken from Nagler et al. (2015).

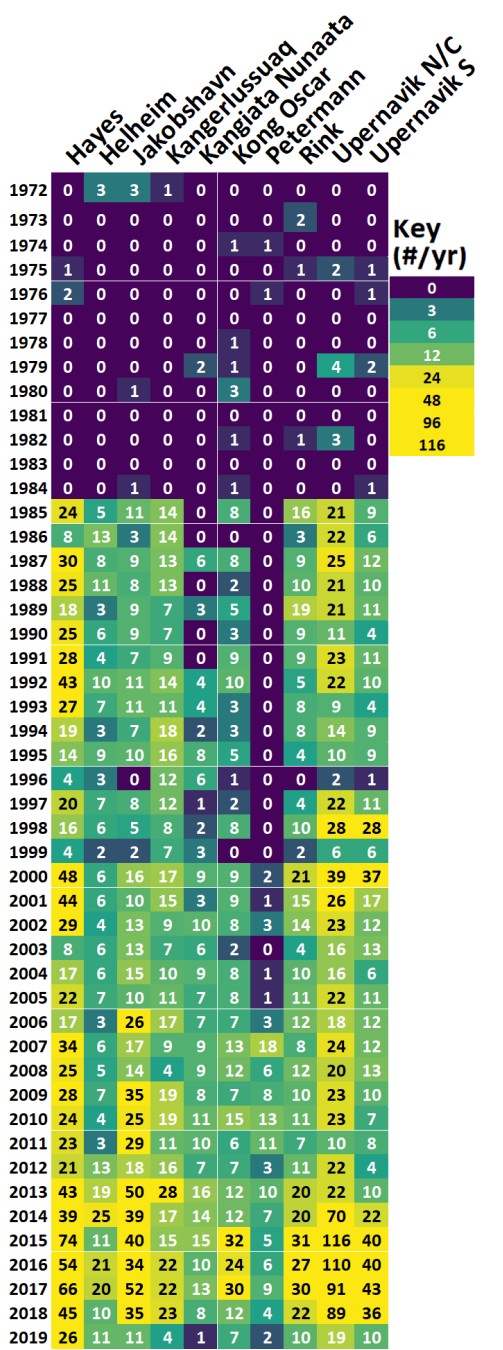

**Figure 2. Temporal Coverage Map**: Number of fronts per year from 1972-2019 for 10 high discharge volume basins. For the full temporal coverage map, see attached Supplement, Fig. S1.

| | Hayes | Helheim | Jakobshavn | Kangerlussuaq | Kangiata Nunaata | Kong Oscar | Petermann | Rink | Upernavik N/C | Upernavik S |
|---|---|---|---|---|---|---|---|---|---|---|
| 1972 | 0 | 3 | 3 | 1 | 0 | 0 | 0 | 0 | 0 | 0 |
| 1973 | 0 | 0 | 0 | 0 | 0 | 0 | 0 | 2 | 0 | 0 |
| 1974 | 0 | 0 | 0 | 0 | 0 | 1 | 1 | 0 | 0 | 0 |
| 1975 | 1 | 0 | 0 | 0 | 0 | 0 | 0 | 1 | 2 | 1 |
| 1976 | 2 | 0 | 0 | 0 | 0 | 0 | 1 | 0 | 0 | 1 |
| 1977 | 0 | 0 | 0 | 0 | 0 | 0 | 0 | 0 | 0 | 0 |
| 1978 | 0 | 0 | 0 | 0 | 0 | 1 | 0 | 0 | 0 | 0 |
| 1979 | 0 | 0 | 0 | 0 | 2 | 1 | 0 | 0 | 4 | 2 |
| 1980 | 0 | 0 | 1 | 0 | 0 | 3 | 0 | 0 | 0 | 0 |
| 1981 | 0 | 0 | 0 | 0 | 0 | 0 | 0 | 0 | 0 | 0 |
| 1982 | 0 | 0 | 0 | 0 | 0 | 1 | 0 | 1 | 3 | 0 |
| 1983 | 0 | 0 | 0 | 0 | 0 | 0 | 0 | 0 | 0 | 0 |
| 1984 | 0 | 0 | 1 | 0 | 0 | 1 | 0 | 0 | 0 | 1 |
| 1985 | 24 | 5 | 11 | 14 | 0 | 8 | 0 | 16 | 21 | 9 |
| 1986 | 8 | 13 | 3 | 14 | 0 | 0 | 0 | 3 | 22 | 6 |
| 1987 | 30 | 8 | 9 | 13 | 6 | 8 | 0 | 9 | 25 | 12 |
| 1988 | 25 | 11 | 8 | 13 | 0 | 2 | 0 | 10 | 21 | 10 |
| 1989 | 18 | 3 | 9 | 7 | 3 | 5 | 0 | 19 | 21 | 11 |
| 1990 | 25 | 6 | 9 | 7 | 0 | 3 | 0 | 9 | 11 | 4 |
| 1991 | 28 | 4 | 7 | 9 | 0 | 9 | 0 | 9 | 23 | 11 |
| 1992 | 43 | 10 | 11 | 14 | 4 | 10 | 0 | 5 | 22 | 10 |
| 1993 | 27 | 7 | 11 | 11 | 4 | 3 | 0 | 8 | 9 | 4 |
| 1994 | 19 | 3 | 7 | 18 | 2 | 3 | 0 | 8 | 14 | 9 |
| 1995 | 14 | 9 | 10 | 16 | 8 | 5 | 0 | 4 | 10 | 9 |
| 1996 | 4 | 3 | 0 | 12 | 6 | 1 | 0 | 0 | 2 | 1 |
| 1997 | 20 | 7 | 8 | 12 | 1 | 2 | 0 | 4 | 22 | 11 |
| 1998 | 16 | 6 | 5 | 8 | 2 | 8 | 0 | 10 | 28 | 28 |
| 1999 | 4 | 2 | 2 | 7 | 3 | 0 | 0 | 2 | 6 | 6 |
| 2000 | 48 | 6 | 16 | 17 | 9 | 9 | 2 | 21 | 39 | 37 |
| 2001 | 44 | 6 | 10 | 15 | 3 | 9 | 1 | 15 | 26 | 17 |
| 2002 | 29 | 4 | 13 | 9 | 10 | 8 | 3 | 14 | 23 | 12 |
| 2003 | 8 | 6 | 13 | 7 | 6 | 2 | 0 | 4 | 16 | 13 |
| 2004 | 17 | 6 | 15 | 10 | 9 | 8 | 1 | 10 | 16 | 6 |
| 2005 | 22 | 7 | 10 | 11 | 7 | 8 | 1 | 11 | 22 | 11 |
| 2006 | 17 | 3 | 26 | 17 | 7 | 7 | 3 | 12 | 18 | 12 |
| 2007 | 34 | 6 | 17 | 9 | 9 | 13 | 18 | 8 | 24 | 12 |
| 2008 | 25 | 5 | 14 | 4 | 9 | 12 | 6 | 12 | 20 | 13 |
| 2009 | 28 | 7 | 35 | 19 | 8 | 7 | 8 | 10 | 23 | 10 |
| 2010 | 24 | 4 | 25 | 19 | 11 | 15 | 13 | 11 | 23 | 7 |
| 2011 | 23 | 3 | 29 | 11 | 10 | 6 | 11 | 7 | 10 | 8 |
| 2012 | 21 | 13 | 18 | 16 | 7 | 7 | 3 | 11 | 22 | 4 |
| 2013 | 43 | 19 | 50 | 28 | 16 | 12 | 10 | 20 | 22 | 10 |
| 2014 | 39 | 25 | 39 | 17 | 14 | 12 | 7 | 20 | 70 | 22 |
| 2015 | 74 | 11 | 40 | 15 | 15 | 32 | 5 | 31 | 116 | 40 |
| 2016 | 54 | 21 | 34 | 22 | 10 | 24 | 6 | 27 | 110 | 40 |
| 2017 | 66 | 20 | 52 | 22 | 13 | 30 | 9 | 30 | 91 | 43 |
| 2018 | 45 | 10 | 35 | 23 | 8 | 12 | 4 | 22 | 89 | 36 |
| 2019 | 26 | 11 | 11 | 4 | 1 | 7 | 2 | 10 | 19 | 10 |

Key (#/yr): 0, 3, 6, 12, 24, 48, 96, 116

## 3 Methods

The automated data processing methodology uses innovative techniques and state-of-the-art neural networks to process raw Landsat and Sentinel 1A/B data into useful calving front Shapefiles. The following section explores this methodology, as outlined by the flowchart below (Fig. 3).

### 3.1 Preprocessing

The first stage involves preprocessing the input data for use with the neural network, as illustrated in Fig. 4. The proceeding steps cover the details of handling Landsat data, but can be applied to Sentinel 1 data for validation purposes. To begin, raster images are selected from areas centered around one of 9 primary glacial basins. These basins include Kong Oscar, Hayes, Rink Isbrae, Upernavik, Jakobshavn, Kangiata Nunaata, Helheim, Kangerlussuaq, and Petermann. Next, all L1TP (precision and terrain corrected) rasters from Landsats 1-8 with low cloud coverage (<20%) are collected. A few L1GS/L1GT (non-corrected) products are also selected, which are manually georeferenced, and used to fill in Landsat 1-2 time series gaps (1972-1985). This results in a total of 4956 Landsat rasters. Next, predefined basin domain Shapefiles that enclose the terminus are used to clip the Landsat raster subsets. Additional filtering removes subsets that still contain ≥30% NODATA pixels or ≥20% cloud pixels detected in the Landsat QA band, as subsets that exceed these thresholds are not likely to contain detectable fronts. At this stage, 20188 GeoTIFF subsets are accumulated. Each subset is then resized to 256x256 px, and lastly enhanced using Pseudo-HDR Toning (HDR) and Shadows/Highlights (S/H) through Adobe Photoshop. The raw, HDR, and S/H enhanced subsets are then stacked into a single RGB image. At this point, the images are ready for processing into calving front masks.

### 3.2 Neural Network Processing

Images are processed using the Calving Front Machine Neural Network (CALFIN-NN), as illustrated in Fig. 5. Neural networks like CALFIN-NN work by learning patterns in training data, and finding them in new data. CALFIN-NN is trained using manually delineated calving front masks. Once trained, CALFIN-NN outputs a probability mask that shows each pixel's likelihood of lying on the coastline/calving front. CALFIN-NN also generates a ice/ocean probability mask as a secondary output. Following this, the calving front is extracted during post-processing, discussed in Sect. 3.3.

Neural networks are the foundation of several automated delineation methods, including Mohajerani et al. (2019), Zhang et al. (2019), and Baumhoer et al. (2019). This method builds upon this work, and uses a modification of the DeepLabV3+ Xception neural network from Chen et al. (2018), as shown in Fig. 5. The first half, the encoder, uses the Xception-65 network to extract image features (Chollet, 2017). It does this by assembling basic features, like edges and corners, into more abstract features, such as glacier/land textures. The second half of the network, the decoder, takes the output of the encoder and up-samples the features to predict the final probability mask outputs.

Several architectural modifications are made to the original DeepLabV3+ Xception model to enhance its perfor-

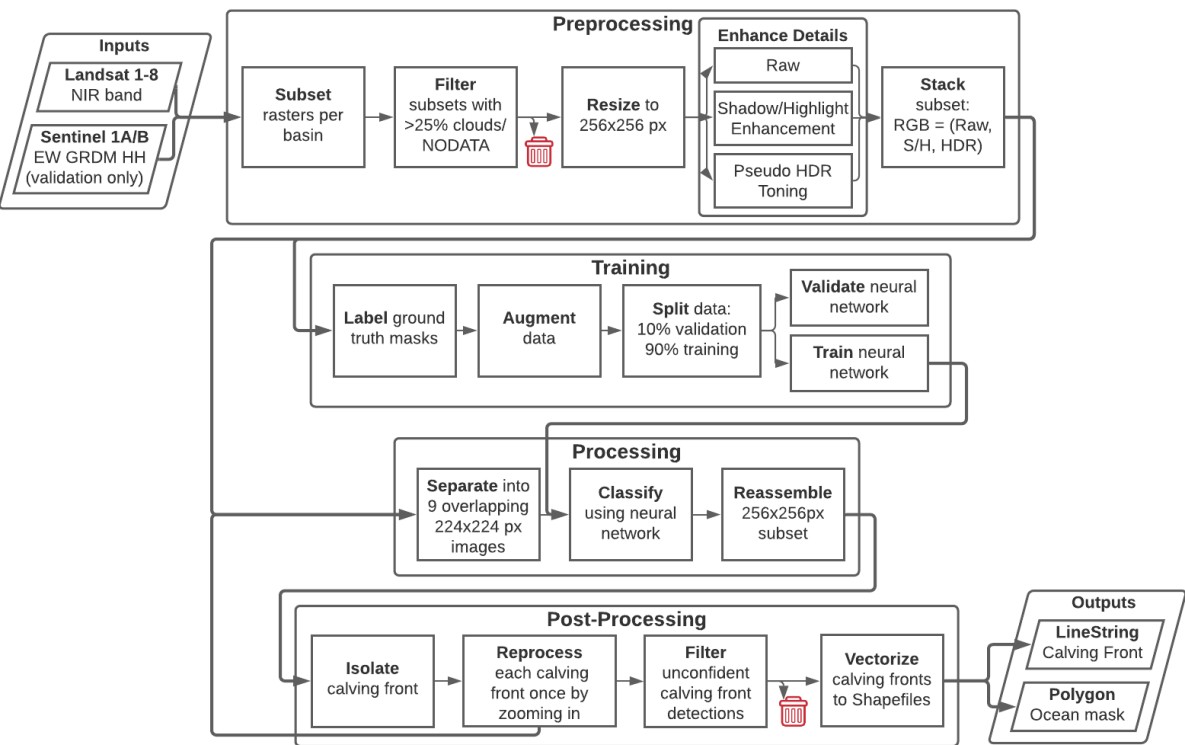

**Figure 3. Methodology Flowchart**: The CALFIN workflow, which processes single band raster imagery into calving front and ocean mask Shapefiles. Note that Sentinel 1A/B imagery is only used for validation, as it is not corrected and thus not qualified for geolocation/extraction.

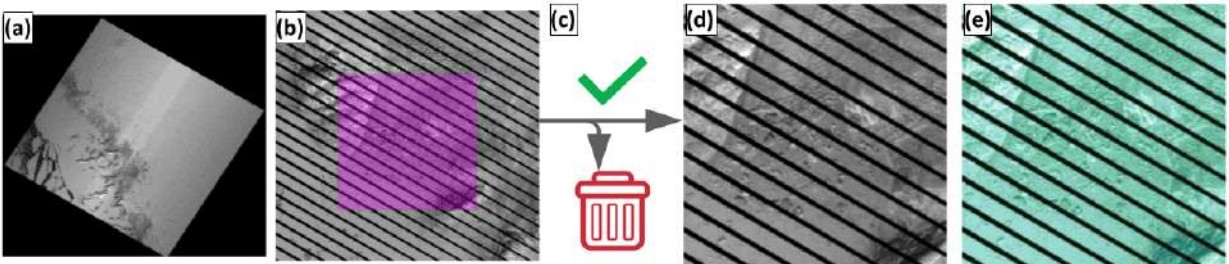

**Figure 4. Preprocessing Pipeline**: (a) First, input the raw Landsat GeoTIFF rasters with <20% clouds. (b) Next, subset using QGIS/GDAL and the domain Shapefile to clip each raster. (c) Then, filter the clouded/NODATA subsets. (d) Now, resize the subsets to 256x256 px. (e) Finally, enhance contrast and stack with the raw subset.

mance. To accurately recognize line-like features such as calving fronts, additional Atrous Spatial Pyramidal Pooling (ASPP) blocks are added in between the encoder and decoder, with the dilation scales 0, 1, 2, 3, 4, and 5. The number of Middle Blocks (MB in Fig. 5) is reduced from 16 to 8, as the extra discriminative power from those blocks is not needed. The input size is reduced from 512 px to 224 px to facilitate better computational performance, allowing for additional training and thus higher accuracy. Since the input resolution is reduced, the encoder is also modified to remove several down-sampling "max-pool" layers. The last contribution adds a 2-channel output to the decoder, allowing for both calving front masking and ice/ocean masking. Together,

these changes reduce number of model parameters from 40M to 29M, while also increasing the overall accuracy.

Several techniques are used during the training of CALFIN-NN to improve its performance. First, a large set of training data is manually delineated (see Fig. S4), totalling 1541 Landsat and 232 Antarctic Sentinel 1A/B image/mask pairs, with the Antarctic data taken from the same training scenes used by Baumhoer et al. (2019). Data augmentation is used to increase the accuracy of the network by expanding the training set, which entails adding random amounts of flips, Gaussian noise, sharpening filters, rotations of up to 12°, crops, and scaling to the pre-processed training images. Through empirical testing, it is determined that excessive image padding, rotation, warping, and cropping of calv-

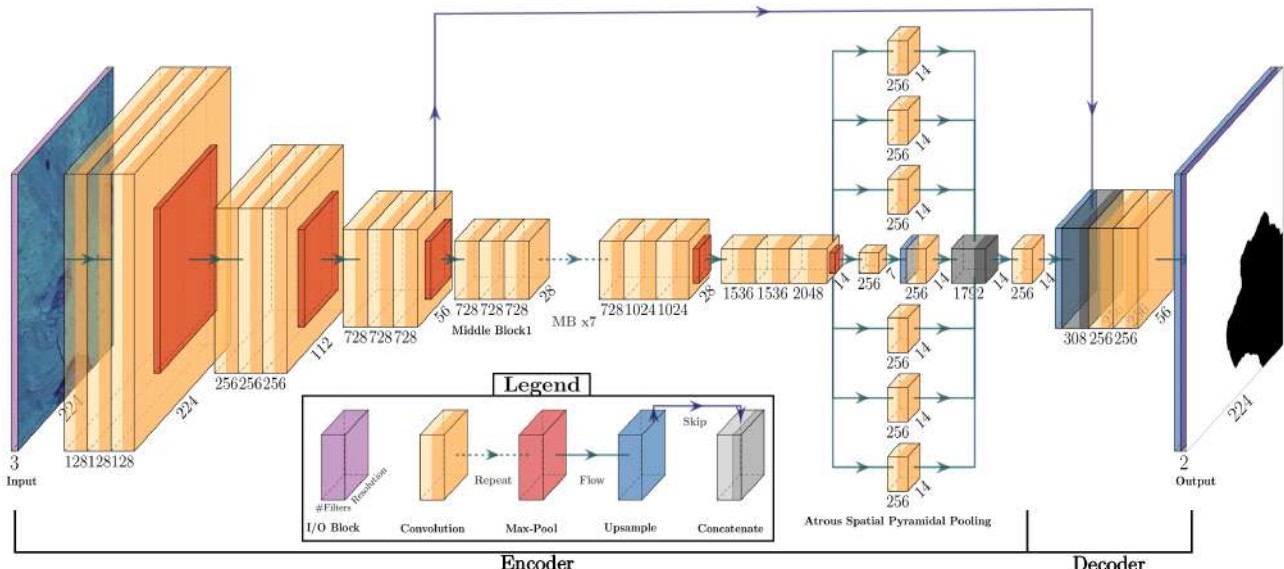

**Figure 5. The CALFIN-NN Processing Architecture**: Each orange "Xception" block consists of convolution kernels that detect features in the previous block. Blocks are reduced in size periodically to pool increasingly complex and numerous feature maps. "U" shaped connections help refine the probability masks during up-sampling. Note that the 7 repeated "Xception" blocks in the middle section are omitted for brevity.

ing fronts to close to the image bounds result in sub-optimal performance. Another helpful technique is the use of test-time augmentations, wherein each image subset is cut into 9 overlapping 224x224 image windows and processed individually, before being reassembled into the final 256x256 output mask. This allows for multiple independent classifications of the central pixels, ensuring agreement and confidence in detected calving fronts. To increase accuracy, a custom loss function optimizes the binary cross entropy and Intersection-over-Union (see Eq. 1, Sect. 4.1) (Mannor et al., 2005). This penalizes mismatches between calving front pixels in the predicted ($\mathbf{I_{cf}}$) and measured ($\hat{\mathbf{I}}_{\mathbf{cf}}$) image masks. Mismatched ice/ocean pixels in the predicted ($\mathbf{I_{io}}$) and measured ($\hat{\mathbf{I}}_{\mathbf{io}}$) image masks are less heavily weighted by an empirically chosen factor of $\alpha = 1/25$, as seen in the final loss function $\mathcal{L}$ in Eq. 2.

$$\mathcal{L}_{\mathcal{BI}}(\mathbf{I}, \hat{\mathbf{I}}) = -\mathbf{I}\log(\hat{\mathbf{I}}) - (1 - \mathbf{I})\log(1 - \hat{\mathbf{I}}) - \log\left(\frac{\mathbf{I} \cap \hat{\mathbf{I}}}{\mathbf{I} \cup \hat{\mathbf{I}}}\right)$$
(1)

$$\mathcal{L}(\mathbf{I_{cf}}, \hat{\mathbf{I}}_{\mathbf{cf}}, \mathbf{I_{io}}, \hat{\mathbf{I}}_{\mathbf{io}}) = \alpha\mathcal{L}_{\mathcal{BI}}(\mathbf{I_{io}}, \mathbf{I_{io}}) + (1 - \alpha)\mathcal{L}_{\mathcal{BI}}(\mathbf{I_{cf}}, \hat{\mathbf{I}}_{\mathbf{cf}})$$
(2)

After integrating these improvements, CALFIN-NN is trained for a total of 80 epochs, with 4000 batches per epoch, and 8 images per batch. Training is carried out on a K40 Nvidia Tesla GPU with 12GB of VRAM, with each epoch taking about 126 minutes to complete, and almost 1 week in total to obtain the optimal weights at epoch 65. Once trained, an NVIDIA GTX1060 with 6GB VRAM is used for the off-line data processing of the 20188 GeoTIFF subsets.

The CALFIN algorithm takes about 3.5 days to process all of the subsets into calving fronts, excluding preprocessing, but including post-processing, as discussed in the following section.

### 3.3 Post-Processing

At this stage, the 2-channel pixel mask output of CALFIN-NN is post-processed to extract the Shapefile data products (Fig. 6).

First, a polyline is fit to the pixel mask to retrieve the correct coastline boundary. This is performed by converting each pixel in the mask to nodes in a graph, connecting the nearest neighboring nodes, then finding the single longest path in the graph's minimum spanning tree (MST) (Kruskal, 1956). This path not only corresponds with the coastline edge, but also out-performs outputs from other contour finding algorithms by eliminating noise, errors, and gaps inherited from previous steps. Such gaps are given weights based on the negative exponential distances between nodes, which allows for connections if the joined paths are significantly longer than the gap itself. A visual example is given in Fig. 7a-d.

Next, the calving front is isolated from the coastline polyline. Static masks of the average fjord boundaries are manually created for each basin using the image subsets and Bed-Machine v3 for reference (Morlighem et al., 2017). By calculating the distance from each point in the coastline to the nearest fjord boundary pixel, then selecting the contiguous pixels which are the farthest from the fjord boundaries, the calving front can be isolated. The result of this is shown in Fig 7e.

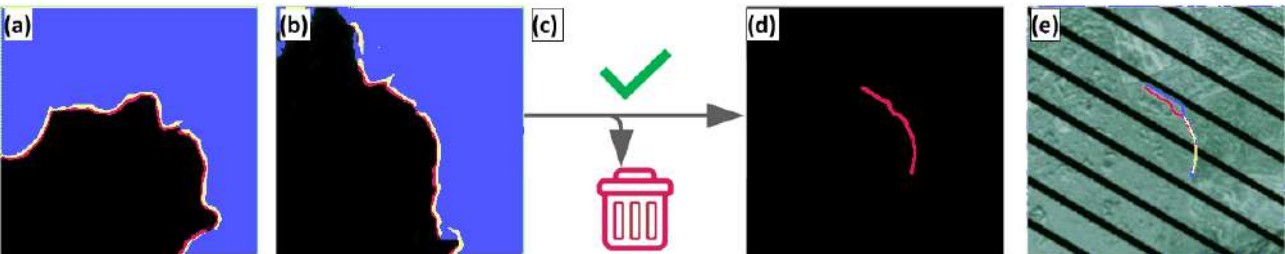

**Figure 6. Postprocessing Pipeline**: (a) First, get the processed image from CALFIN-NN. (b) Then, isolate and re-process each front. (c) Next, filter unconfident predictions. (d) Now, fit line and mask static coastline (see also Fig. 7). (e) Lastly, export and validate the Shapefile.

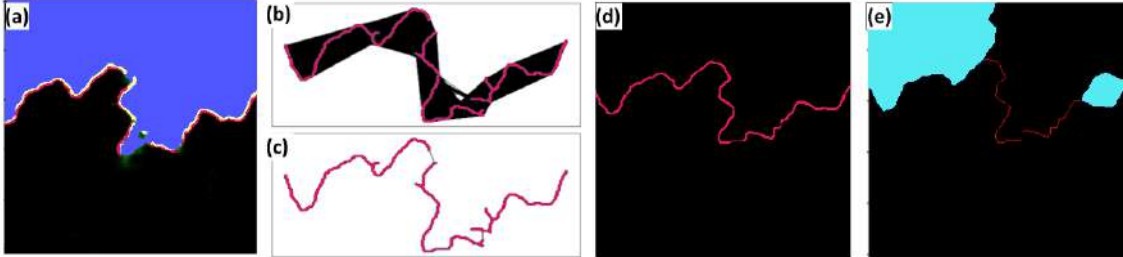

**Figure 7. Mask to Polyline Algorithm**: (a) First, extract the coastline mask (red/yellow) from the CALFIN-NN output. (b) Then create a graph, connecting each pixel (red) to 15% of its nearest neighbors with an edge (black). (c) Next, create an MST from the graph. (d) Now, extract the longest path from the MST. (e) Finally, mask the static coastline using the fjord boundaries (cyan) to extract the calving front.

Once each front is located, its bounding box is used to extract a higher resolution subset from the original image, and reprocessed. This innovation allows for increased spatial accuracy when processing multiple fronts in large basins. After reprocessing, the nature of CALFIN-NN's 2-channel output as a confidence measure is exploited to filter out uncertain detections. Since the neural network assigns each pixel a value between 0 and 1 based on its perceived class, any deviation from these two values can used as a measure of uncertainty. The filtering method averages the deviation of the ice/ocean classification mask in a 5 pixel wide buffer around the calving front, and discards any fronts whose mean deviation exceeds an empirically chosen threshold of 0.125.

The last step is to export the polylines and the corresponding polygon as geo-referenced Shapefiles. First, the polylines are smoothed to eliminate noise artifacts inherited from previous steps, deviating no more than 1 pixel from the raw extracted coastline (see Supplement Fig. S2). Next, the smoothed polylines, fjord boundary mask, and land-ice/ocean masks are combined to create a polygonal ocean mask. Optionally, manual verification of each output with the original GeoTIFF subset can be performed. This was done for all cases in this study to ensure the validity of the automated pipeline. This constrains the mean distance error to be <100 m, as covered in the following section.

## 4  Validation

Two methods are used to evaluate CALFIN. For the primary method, the error is estimated by calculating the Mean/Median Distance between predicted and manually delineated fronts (see Fig. 8a and Sect. 4.1). For the secondary method, the classification accuracy is calculated with the Intersection over Union metric (see Fig. 8b and Sect. 4.2). Additionally, the detection accuracy is evaluated, and the associated confusion matrix is provided (see Table 1 and Sect. 4.4). These metrics are evaluated on several validation sets, taken from existing studies as discussed in Sect. 1. These validation sets contain data that are excluded during model training. This prevents the models from memorizing data and skewing the accuracy assessment.

### 4.1  Error Estimation

The primary quality assessment method is the Mean Distance Error (Mohajerani et al., 2019; Zhang et al., 2019; Baumhoer et al., 2019). Conceptually, this method resembles the numerical integration of the area between two curves, normalized by the average length of the curves (see Fig. 8a). Also referred to as the Area over Front (A/F) in literature, this method can also be seen as a generalization of the method of transects along arbitrarily oriented fronts (Mohajerani et al., 2019; Baumhoer et al., 2019). This metric is implemented by taking the mean/median of the distances between closest pixels in the predicted and manually delineated fronts. Note that pixel distance is biased to be inversely proportional to a

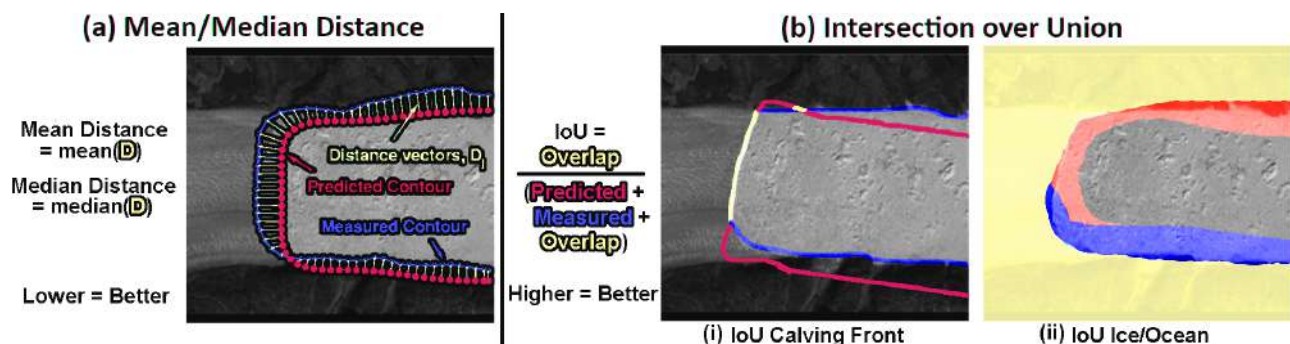

**Figure 8. Error Measures**: (a) A visual outline of Mean/Median Distance Error Estimation and (b) Classification Accuracy using Intersection over Union (IoU) for (i) the primary calving front, and (ii) the secondary ice/ocean mask, respectively.

network's input size, so the error in meters is also provided in the following analysis.

## 4.2 Classification Accuracy

The secondary quality assessment method calculates the Intersection over Union (IoU) (Baumhoer et al., 2019). This metric evaluates the degree of overlap between the predicted and manually delineated masks of the calving front. It is calculated by dividing the number of pixels in the intersection of two masks over the number of pixels in the union of the two masks (see Fig. 8b). When calculating the IoU of 3 pixel wide edges, this measure is very strict: 1 pixel of difference results in a score of 0.5, and scores at or above that range are indicative of human levels of accuracy. When calculating the IoU of land-ice/ocean masks, this measure is less strict, and scores at or above 0.9 indicate human levels of accuracy.

## 4.3 Validation Results

The following subsections show tables with the above metrics for the associated validation sets, the values from the original studies, and a subset of the outputs of CALFIN-NN on each. The primary validation set, the CALFIN validation set (CALFIN-VS), consists of 162 images with clouds, illumination differences, ice mélange, and Landsat 7 Scanline Corrector Errors (L7SCEs). The CALFIN-VS contains data from 62 Greenlandic basins, including Helheim, which was specifically excluded from CALFIN's training set for validation purposes - as done by Mohajerani et al. (2019). The CALFIN-VS ensures CALFIN-NN produces consistent results on new data, addressing concerns raised by Zhang et al. (2019) Sect. 7.3. To evaluate performance on Landsat 7 Scanline Corrector Errors, the validation subset CALFIN-VS-L7-only isolates images with L7SCEs, and the CALFIN-VS-L7-none excludes images with L7SCEs. To allow for comparisons between studies, CALFIN-NN's performance metrics on previous studies' validation sets are also shown, where appropriate. The sets include the 10 Landsat Helheim subsets used in Mohajerani et al. (2019) (M-VS), the 6 TerraSAR-X Jakobshavn subsets used in Zhang et al. (2019) (Z-VS), and

62 Sentinel-1 Antarctic basins taken from the 11 validation scenes used in Baumhoer et al. (2019) (B-VS). Note that the error metrics are still sensitive to how each study implements them, which are nevertheless reproduced and documented for comparison's sake. These concerns are also addressed in the comprehensive inter-model comparison, discussed in Sect. 6.

CALFIN-NN performs well on the CALFIN-VS (Fig. 9). The true mean distance error of the CALFIN dataset is calculated to be $86.76 \pm 1.43$ m with 95% confidence. When including only images with L7SCEs (CALFIN-VS-L7-only), the error is 91.93 m, showcasing CALFIN-NN's unique robustness to L7SCEs. Intuitively, excluding "difficult" images with L7SCEs in the validation set (CALFIN-VS-L7-none) decreases the error to 81.65 m. The median distance error is only 44.59 m, showing that only a few outliers contribute considerably to the mean. For full outputs, see Supplement Figs. S5-S8.

CALFIN-NN performs well on the M-VS (Fig. 10). This demonstrates CALFIN-NN's ability to accurately process new data, which builds upon the Mohajerani et al. (2019) neural network (M-NN). Note that M-NN implements distances errors differently, and omits ice/ocean masks from the evaluation. This differences are further explored in the Sect. 6 model inter-comparison.

CALFIN-NN performs competitively on the Z-VS (Fig. 11). It achieves a similar mean meter distance (115.24 m vs. 104 m) despite being constrained to using lower resolution TerraSAR-X data. Note though that the Zhang et al. (2019) neural network (Z-NN) uses higher resolution input data (960×720) compared to CALFIN-NN (224x224), which skews the mean pixel distance comparison, where CALFIN-NN performs better (2.11 px vs. 17.3 px). Another source of skew comes from CALFIN-NN confidence filtering, as only 8 of 12 fronts in the set are confidently detected (see Sect. 4.4). Increasing CALFIN-NN's input resolution and training on higher resolution SAR data may enable CALFIN-NN to detect more fronts with greater accuracy.

CALFIN-NN performs sub-par on the B-VS (Fig. 12). When comparing the mean distance error with the Baumhoer et al. (2019) equivalent Area over Front (A/F) error, the

| Validation Set | Model | Mean Distance | Median Distance | IoU Calving Front | IoU Ice/Ocean |
|---|---|---|---|---|---|
| CALFIN-VS | CALFIN-NN | 2.25 px, 86.76 m | 1.21 px, 44.59 m | 0.4884 | 0.9793 |
| CALFIN-VS-L7-none | CALFIN-NN | 2.27 px, 81.65 m | 1.16 px, 44.01 m | 0.4880 | 0.9819 |
| CALFIN-VS-L7-only | CALFIN-NN | 2.22 px, 91.93 m | 1.33 px, 49.24 m | 0.4888 | 0.9766 |

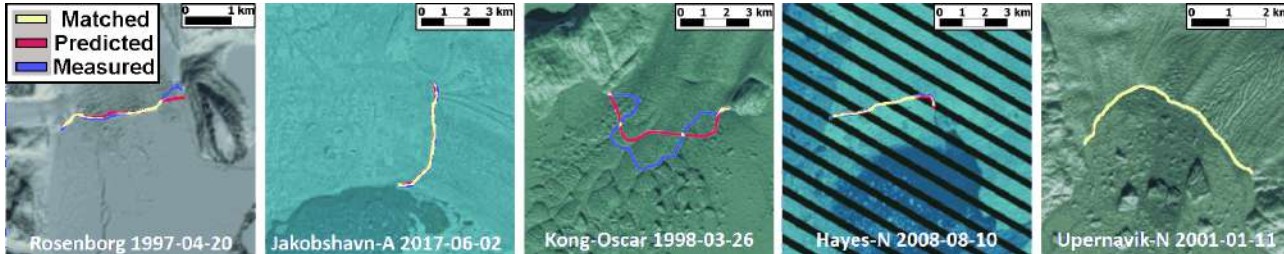

**Figure 9. CALFIN-VS Validation Output Results**: Yellow represents human (green) and machine (red) agreement on the front location. Note that the drop in mean pixel distance despite the increase in mean meter distance (and vice versa) comes from L7SCE images being reprocessed at lower sizes due to detection failures (see Fig. 6c), and pixel error bias being inversely related to input size (see Sect. 4.1).

| Validation Set | Model | Mean Distance | Median Distance | IoU Calving Front | IoU Ice/Ocean |
|---|---|---|---|---|---|
| M-VS | CALFIN-NN | 2.56 px, 97.72 m | 2.55 px, 97.44 m | 0.3332 | N/A |
| M-VS | M-NN | 1.97 px, 96.31 m | N/A | N/A | N/A |

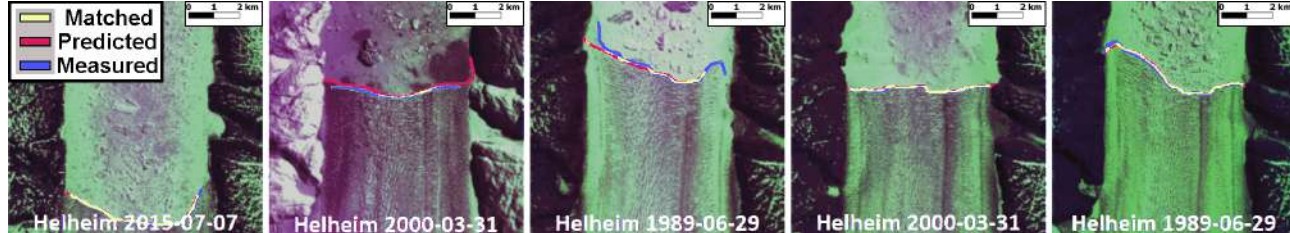

**Figure 10. M-VS Validation Output Results**: Note that CALFIN-NN has never trained on Helheim, but can still predict the front under different conditions and preprocessing methods. See Fig. S9. for full outputs.

| Validation Set | Model | Mean Distance | Median Distance | IoU Calving Front | IoU Ice/Ocean |
|---|---|---|---|---|---|
| Z-VS | CALFIN-NN | 2.11 px, 115.24 m | 1.65 px, 77.29 m | 0.3832 | 0.9761 |
| Z-VS | Z-NN | 17.3 px, 104 m | N/A | N/A | N/A |

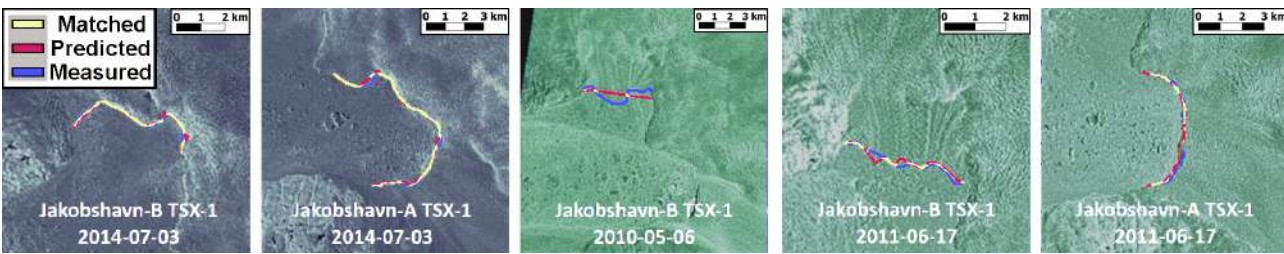

**Figure 11. Z-VS Validation Output Results**: CALFIN-NN works well on SAR data in addition to optical data. See Fig. S10. for full outputs.

| Validation Set | Model | Mean Distance | Median Distance | IoU Calving Front | IoU Ice/Ocean |
|---|---|---|---|---|---|
| B-VS | CALFIN-NN | 2.35 px, 330.63 m | 0.74 px, 112.75 m | 0.6451 | 0.9879 |
| B-VS | B-NN | 2.69 px, 108 m | N/A | N/A | 0.905 |

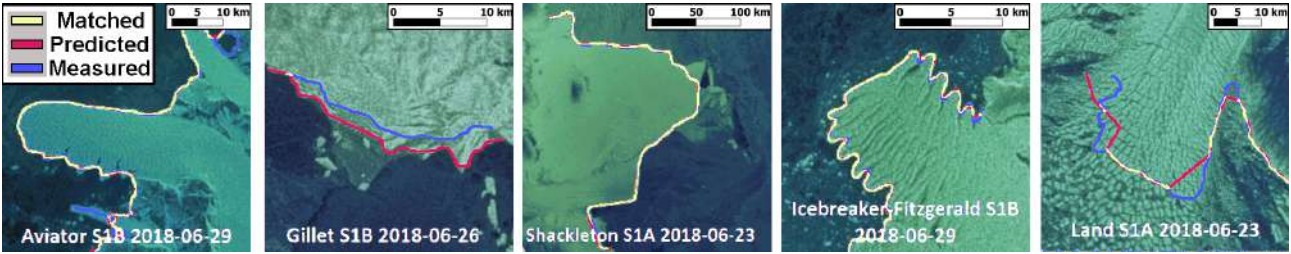

**Figure 12. B-VS Validation Output Results**: Similar to Z-NN, B-NN uses a high resolution input (768×768) relative to CALFIN-NN (224x224), which skews the mean pixel distance comparison in CALFIN-NN's favor. See Fig. S11-S12 for full outputs.

Baumhoer et al. (2019) neural network (B-NN) outperforms CALFIN-NN (330.63 m vs 108 m). Note that the easily detected static coastlines are masked out, raising the relative error, and negatively impacting CALFIN-NN's performance on this metric. When comparing metrics that isolate the calving front, the absolute median distance error is calculated (achieving 112.75 m) whereas Baumhoer et al. (2019) uses signed median distance error (achieving 0 m), which is not directly comparable in this context, and thus omitted. Currently, the error is affected by kilometer-range deviations in very large domains like Voyeykov Ice Shelf, and differences in sea-ice mélange as seen along the Gillet and Wordie Ice Shelves, which would be consistent with findings in Baumhoer et al. (2019) Sect. 5.2. After excluding such outliers, fronts are detected in 55 out of 62 domains (88.71%), achieving median distance errors of 0.95 px (127.87 m). Intensive retraining on ice shelves may be required for CALFIN-NN to improve.

### 4.4 Detection Accuracy

Lastly, CALFIN-NN is shown to automatically filter images that do not have detectable calving fronts. To verify this, 13 images are included in the CALFIN-VS which do not contain calving fronts discernible to the human eye. The true positive (TP), true negative (TN), false positive (FP), and false negative (FN) rates are computed for the entire 162 image CALFIN-VS, and the associated confusion matrix is shown in Table 1. Note that CALFIN-NN does not output any false positives on the CALFIN-VS. While this ensures accurate fronts are output rather than incorrect fronts, this filtering behavior removes potentially large errors, and must be accounted for when comparing errors across other sets.

**Table 1. Confusion Matrix**: CALFIN-NN misses fronts in 8 of 149 valid CALFIN-VS images, but this trade-off is acceptable.

|  |  | Front Detected? | |
|---|---|---|---|
|  |  | Yes | No |
| Front Detectable? | Yes | TP = 141/149 (94.63%) | FN = 8/149 (5.76%) |
|  | No | FP = 0/13 (0.00%) | TN = 13/13 (100.00%) |

## 5   Results and Discussion

The code implementation of the CALFIN method is released, along with its associated calving front data products as described in the following section, for use within the scientific community. The CALFIN dataset spans 66 Greenlandic basins, over the period Sept. 1972 - June 2019. It consists of over 1500 manual delineations and 22,678 total calving fronts. Two levels of CALFIN data products are provided. The Level 0 products include the Shapefile domains used for subsetting, the neural network training image/mask pairs, the fjord boundary masks, the full Landsat scene ID list, and the quality assurance images for validation purposes. The use cases of Level 0 products may include studies of reproducibility, validation, or training new neural networks. The Level 1 products include the calving front polyline and polygon Shapefiles. The polyline product consists of the isolated, refined, geo-referenced, and verified calving fronts for each domain. The polygon product consists of an ocean mask bounded by the domain subset, the fjord boundaries, and the calving front(s), for each domain. Both of the Shapefiles share a common metadata feature schema (see Table S2) derived from the MEaSUREs Glacial Termini Dataset (Moon and Joughin, 2008; Joughin et al., 2015), and names are derived from Bjørk et al. (2015). These products can be found via these links to Github and DataDryad (Cheng et al., 2020).

With the new data available to use in the CALFIN dataset, it is possible to explore seasonal trends across the Greenland Ice Sheet, and validate a subset of 10 high discharge basins of interest against existing ESA-CCI, MEaSUREs, and PROMICE data products (ENVEO, 2017; Joughin et al., 2015; Andersen et al., 2019). Fig 13 shows the high temporal resolution and spatial accuracy of the CALFIN data product alongside corresponding available data products from 1972-2019. For Joughin et al. (2015), if a date range is given, the same relative change at both start and end dates (Moon and Joughin, 2008) is plotted. For Andersen et al. (2019), August 15th is used as the "end-of-melt-season" date of delineation, as the date is otherwise not specified in the provided data. The advance and retreat of the calving front along the basin centerlines is relative to their earliest positions. Note the large improvement in temporal/seasonal coverage and the general agreement of CALFIN with existing data products.

# Relative Advance and Retreat, 1972-2019

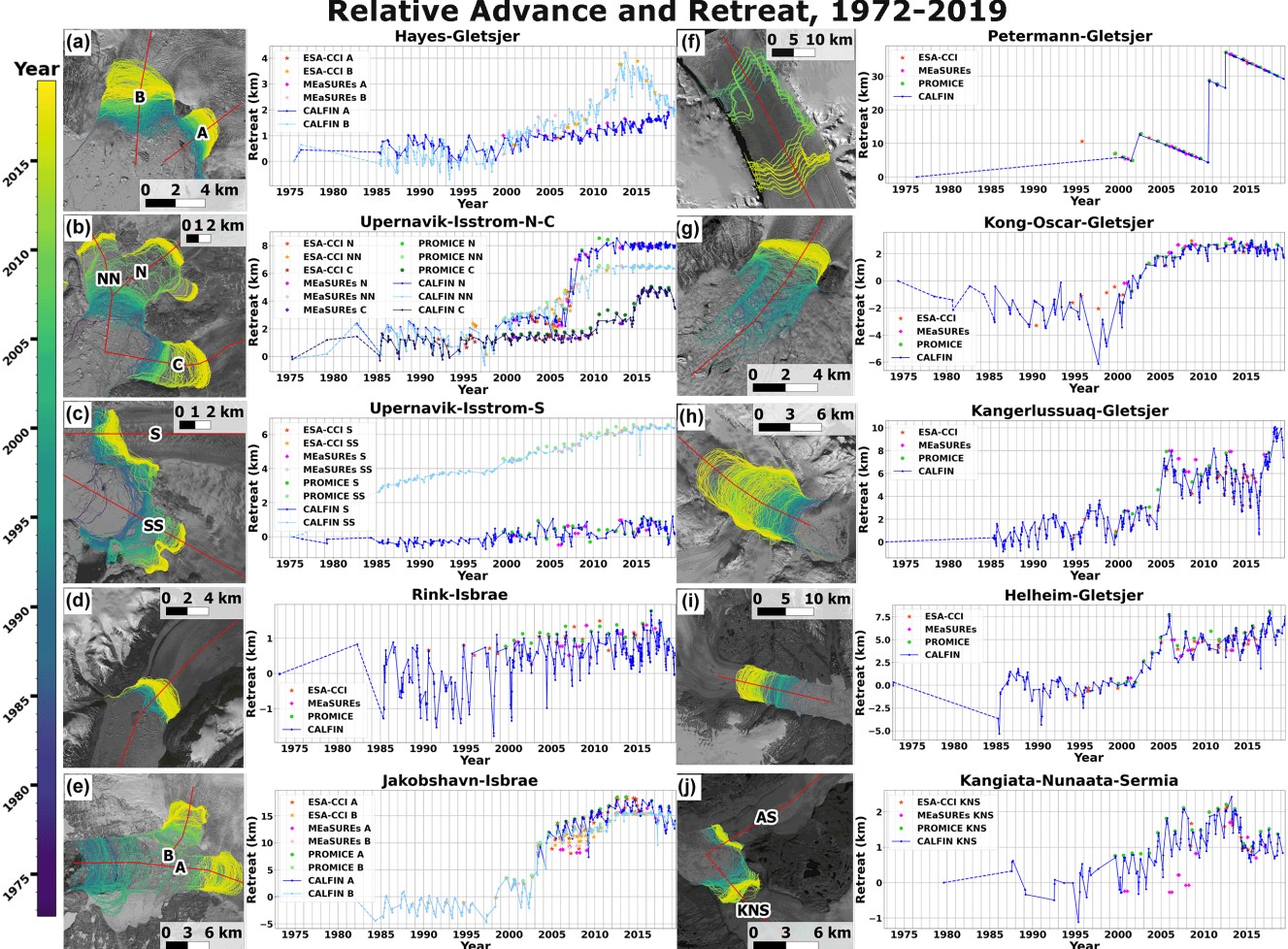

**Figure 13. Terminus Advance and Retreat Over Time**. (a-j) Basin setup (left) and graph (right) for 10 high discharge basins. Positive length change represents retreat relative to the earliest position along the centerlines in red. Note the seasonal variations captured by CALFIN, in blue. Time series for other studies span 1990-2016 (ESA-CCI), 2000-2017 (MEaSUREs), and 1999-2019 (PROMICE). Note the seasonal variations shown by the solid lines, and the dotted lines from 1972-1985 that indicate a lack of such seasonal observations. Also note that the vertical axis scaling is applied differently for each graph to highlight seasonal trends.

Note also that the discrepancies such as that during 2005-2009 in Jakobshavn (Fig. 13e) mostly stem from a lack of winter coverage during Landsat's optical blackout period. Additional outliers in Kong Oscar (Fig. 13g) stem from the somewhat arbitrary delineation of the ice tongue front position. Kangiata Nunaata (Fig. 13j) suffers from both of the aforementioned effects, but otherwise shows the same general agreement with existing datasets from 2000 onwards.

Additionally, Fig. 14 shows the regional mean advance and retreat change, alongside the mean for the entirety of Greenland covered by the CALFIN dataset. Contributions from NW Greenland influence the overall trend the most, due to the presence of many small glaciers/branches in the region. Note that the mean for Greenland also includes contributions from Petermann, which is visible in the summers of 2010 and 2012. Shared regional trends are visible across NW and CW Greenland, which both show relative stabil-

ity before 2000, followed by steady retreat up until 2017-2018. CE and SE Greenland also share a similar but less pronounced retreat, showing an accelerating retreat beginning around 1995. These regional trends are less visible in SW Greenland, which is dominated by Narsap Sermia's retreat from 2010-2013. Overall, these regional trends generally agree with studies such as Wood et al. (2021) and King et al. (2020), helping further validate the CALFIN method and data.

## 6   Inter-model Comparison

To further reinforce the validity of the study, and address the shortcomings of different error metric comparisons (as discussed in Sect. 4.3), a comprehensive inter-model comparison is conducted between CALFIN-NN and the model devel-

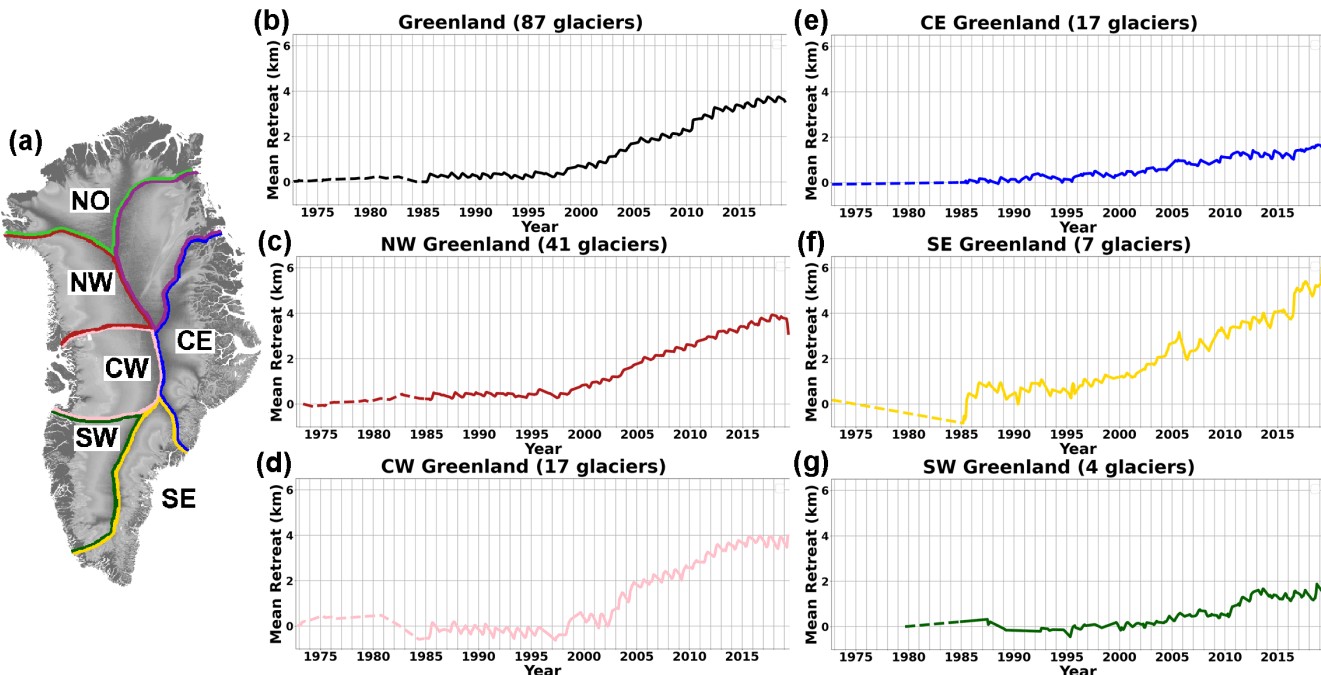

**Figure 14. Regional Terminus Advance and Retreat Over Time**. (a) Regional delineations (left) and terminus position graphs (right) for Greenland (b), as well as the northwestern (c), central western (d), central eastern (e), southeastern (f), and southwestern (g) regions. Note that the total Greenland mean advance and retreat is unadjusted, and dominated by the trend lines of numerous smaller glaciers in CW and NW Greenland. Note that branches in the 66 studied basins are independently counted, for a total of 87 glaciers.

oped by Mohajerani et al. (2019) (M-NN). This experiment seeks to understand how both models perform, holding all other variables constant. In particular, this experiment seeks to determine if the M-NN model, and by extension other UNet models, perform on par with the CALFIN-NN model, given the same training data. This task involves retraining the M-NN on CALFIN training data, and comparing its performance against CALFIN-NN using a shared validation set. For the fairest results, only images without L7SCEs are evaluated in this validation set - CALFIN-VS-L7-none - which is within the known capabilities of the M-NN. Furthermore, the same pre- and post-processing is applied to both models.

Across all non-Landsat 7 test images in the CALFIN validation set, CALFIN-NN attains a 2.27 pixel (81.65 meter) mean distance between the predicted and the manually delineated fronts. This exceeds the level of accuracy achieved by the model from Mohajerani et al. (2019), which after retraining on CALFIN training data, is 4.45 pixels (201.35 meters). Note again that Landsat 7 images were excluded during reevaluation for the M-NN. This supports the findings that the CALFIN-NN architecture is an improvement over existing UNet models.

With this added context, the validation table is reproduced from Sect. 4.3, Fig. 10, and the error analysis is continued below. To reemphasize the differences in mean distance error calculation between different studies, Mohajerani et al. (2019) begins by breaking each predicted front to 1000

smaller segments within a small buffer from the fjord walls and calculating the mean deviation between the segments of the predicted and manually delineated fronts. The method begins by averaging the mean distance between each pixel of the predicted front and the closest pixel of the manually delineated front as detailed in Sect 4.1. While the line-segment methodology of Mohajerani et al. (2019) provides a stricter estimate by enforcing close agreement between corresponding front segments, the CALFIN method allows for non-aligned evaluation of the mean distance error. Although both implementations quantify the differences between the lines, the differences in implementation should still be considered when evaluating the comparison below.

Across all 10 test images in the M-VS, CALFIN-NN attains a 2.56 pixel (97.72 meter) mean distance between the predicted and the manually delineated fronts. This approaches the level of accuracy achieved in the original study, which is 1.97 pixels (96.31 meters). This supports the findings that the CALFIN-NN architecture generalizes to new data well. Note that CALFIN-NN's larger network size requires additional training data to avoid over-fitting, or memorizing, the training data, which could explain the slightly lesser accuracy when compared to the M-NN. In summary, this comprehensive model inter-comparison supports the hypothesis that the CALFIN-NN model improves on existing studies and is generalizing well.

**Table 2. Model Inter-comparison Error Table**: Metrics for the CALFIN-NN and M-NN models on all non-Landsat 7 test images in the CALFIN validation set.

| Validation Set | Training Set | Model | Mean Distance | Median Distance | IoU Front | IoU Ice/Ocean |
|---|---|---|---|---|---|---|
| CALFIN-VS-L7-none | CALFIN | CALFIN-NN | 2.27 px, 81.65 m | 1.16 px, 44.01 m | 0.4880 | 0.9819 |
| CALFIN-VS-L7-none | CALFIN | M-NN | 4.45 px, 201.35 m | 1.25 px, 50.52 m | 0.4935 | 0.9699 |

**Table 3. M-VS Validation Output Results**: Accuracy and error metrics for the CALFIN-NN and the M-NN models on the M-VS. Again, some metrics are not provided by Mohajerani et al. (2019), so they are omitted from this table.

| Validation Set | Training Set | Model | Mean Distance | Median Distance | IoU Front | IoU Ice/Ocean |
|---|---|---|---|---|---|---|
| M-VS | CALFIN | CALFIN-NN | 2.56 px, 97.72 m | 2.55 px, 97.44 m | 0.3332 | N/A |
| M-VS | Mohajerani | M-NN | 1.97 px, 96.31 m | N/A | N/A | N/A |

## 7   Conclusion

Overall, the goal of automatically delineating calving fronts from satellite imagery is accomplished. The CALFIN method uses the cutting-edge in deep learning architectures, allowing for robustness to minor cloud cover, Landsat 7 Scanline Corrector Errors, and illumination changes. The method is validated through a comprehensive data intercomparison with existing studies, and the results deviate by on average 86.76 ± 1.43 meters from the measured fronts. Regional trends show larger than average absolute retreat in SE Greenland, and new sub-seasonal trends are available for further investigation with the release of the 22,678 calving front lines generated across 66 Greenlandic glaciers. Future work may entail accuracy improvements, expansion of included domains, usage of SAR data sources, and near-real time data products. Within the community, the benefits of standardized training, validation sets, and outputs/metadata are anticipated. The community's development of new automated extraction studies, such as grounding line delineation, iceberg tracking, and sea ice mélange measurements, is also anticipated. A key takeaway is the maturation of neural networks for automated calving front detection. Specifically, a well trained network now approaches human levels of accuracy in picking arbitrary glacial calving fronts. This reinforces existing studies on the viability of the methodology, and paves the way for applications on other data processing tasks. Ultimately, this work showcases the state-of-the-art in automated calving front detection, and provides a new database of glacial termini positions for the cryosphere community.

*Code and data availability.* The code used to automate the implement the CALFIN pipeline is freely available at github.com/daniel-cheng/CALFIN. It is written in Python 3, using the Keras & Tensorflow libraries. The data generated by CALFIN is currently available at datadryad.org/stash/dataset/doi:10.7280/D1FH5D.

**The Supplement related to this article is available online at: https://doi.org/10.5194/tc-0-1-2021-supplement**

*Author contributions.* DC developed the code/model, created the training data, carried out the data processing/error analysis, and wrote the majority of the manuscript. WH provided input on the processing methodology, post-processing algorithms, error analysis, discussion topics, and writing the manuscript. EL provided key direction for the overall study, error analysis, outputs, and writing the manuscript. YM performed the model inter-comparison and assisted with the writing of the manuscript. MW performed the data preprocessing for the model inter-comparison. IV assisted in organizing collaborators and the model inter-comparison. ER contributed suggestions regarding the error analysis and inter-comparison. WH, EL, MW, and YM revised the manuscript and results.

*Competing interests.* The authors declare no competing interests.

*Acknowledgements.* This work was conducted as a collaboration between NASA's Jet Propulsion Laboratory and the University of California, Irvine. The CALFIN neural network architecture implementation is derived from Emil Zakirov's Deeplabv3+ Xception codebase at github.com/bonlime/keras-deeplab-v3-plus (last access: 13 August 2020). We acknowledge the USGS for providing Landsat-1-8 images, the ESA for their Sentinel-1 images, as well as the ESA-CCI, PROMICE, and MEaSUREs programs for providing calving front data used in this study. Additionally, we thank the editors and reviewers for their contributions to the improvement of this manuscript.

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
