# Peer review of "Calving Front Machine (CALFIN): Glacial Termini Dataset and Automated Deep Learning Extraction Method for Greenland, 1972-2019"

_The Cryosphere, 2020_

## Referee Comment (RC1) · Anonymous Referee #1 · 9 Nov 2020

General Comments

This manuscript introduces the novel developed Calving Front Machine "CALFIN" for the automated extraction of Greenlandic calving fronts. This is a major contribution to the field as it replaces time-consuming manual delineated fronts by automatically extracted dense glacier front time series. The CALFIN algorithm was validated extensively against test datasets and results from previous studies through a model inter-comparison. The scientific community will definitely benefit from this development as an automatically derived calving front position data set of 66 Greenlandic glaciers will be released with this publication.

[Figure]

Despite the impressive results and technical details of this manuscript, I have some concerns about the structure of this paper and the (sometimes) very short explanations. However, after re-structuring some parts of the manuscript and adding additional information as indicated below, this paper will present an important contribution to the field.

In my opinion, the abstract should be structured more clearly. For a better understanding, I would recommend to re-order the abstract by using the common schema: 1) Statement of the problem, 2) Research question, 3) Research design, 4) Central results, 5) Brief interpretation of the results, and 6) Outlook/ future use of the data set.

P2L4: The paper introduces a new method and provides an inter-comparison with other studies. For readers not familiar with the studies of Zhang et al, Mohajerani et al. and Baumhoer et al. it would be helpful to have a brief state-of-the-art paragraph reviewing existing calving front extraction methods. For example, P2L4 could be extended and give more insights into the studies used in the inter-comparison as well as the studies of Seale et al. 2011 and similar approaches.

P2L11: In my opinion this section is incomplete. Please mention all potential data sources in Table 1 (add Sentinel-2, Envisat, ERS, Radarsat) and justify why they are not suitable. Another option would be to just focus on Landsat data and remove the incomplete Table 1. Figure 1 is really great so I would try to put the focus on it and highlight the incredible amount of processed data and outline the advantages, data amount, and characteristics of Landsat.

P2L17: The methodology section could give a short overview of the entire workflow from pre-processing to the final extracted calving front by showing a flow chart. This would guide the reader through the methodology part and link the numerous subchapters of section 3. Besides, in my opinion, the training of the network explained in P12L2 should be part of the methodology and not subject to the discussion.

Specific Comments

Interactive
comment

P5 Figure 5c: How does the filtering of unconfident predictions work? Please describe this in the methodology section.

P6L1: Please outline the calving front re-processing in more detail. Does the re-processing allow a higher spatial accuracy when re-processing a part of the image?

P6L16: How much smoothing of the extracted coastline is allowed and can this also decrease accuracy?

P8L1: How did you handle the issue that your network was trained for 3-channel RGB imagery but tested on 1-channel SAR data?

P8L18: What are the characteristics of those outlier glaciers and how many glaciers are defined as "outlier"?

P11L15: The information of this section could also be shifted to methodology. Then rename Chapter 5 to "CALFIN Dataset".

P10L4: But also mention the mean distance which is comparable here.

P10 Figure 11: How did you consider the fact that ice shelves are much bigger than glaciers? For example, in Figure 11 you show the Shackleton ice shelf. It is approx. 200 km wide and if you resample that to 224x224 pixels, one pixel for your validation would be 892 m compared to 40 m pixels in the original study by Baumhoer et al. 2019. How did this influence the validation accuracy? For Zhang et al. you show that the use of higher resolution of TerraSAR-X data does not improve the mean distance accuracy (Figure 10).

P13 Figure 13: Can you explain why the PROMICE data set (2008/2009 and 2010/2011) shows twice a very different front position compared to the CALFIN data set?

P13L13: The model inter-comparison is only discussed for the study of Mohajerani et al. but validations were also done against the data sets of Zhang et al. and Baumhoer
et al., hence those results should also be discussed.

---

## Referee Comment (RC2) · Anonymous Referee #2 · 11 Nov 2020

General Comment Cheng et al. present an automated method for delineating glacier calving fronts – named Calving Front Machine (CALFIN) - based on a deep learning approach, accompanied by a new dataset of Greenland glacier termini. The principal input data are Landsat optical images acquired since 1972. The methodology builds on previous work by Mohajerani et al., Zhang et al., and Baumhoer et al. and uses computing systems, named neural networks, that learn patterns in training data, in order to identify similar patterns (such as glacier termini) in new data. The authors detail the various steps of the processing chain and produce a set of shapefiles, which are evaluated and intercompared with both internal and external (manually) retrieved calving front datasets using different quality metrics. The main outcome is an extensive

dataset covering 66 outlet glaciers around Greenland with in total 22,679 individual calving fronts encompassing the period 1972-2019. The method and new data set reportedly exceeds the accuracy of previous work and approaches human levels of accuracy in delineating glacier termini, the key takeaway being the maturation of neural networks for automated calving front detection.

Automated calving front extraction is a long sought after goal, that recently gained new attention thanks to advances in modern computing technology and increasing availability of satellite EO data. The use of deep learning/neural networks – the subject of this paper - to achieve this is very promising indeed. This paper by Cheng et al. is a welcome addition to existing literature on this topic as is the associated dataset for the community, expanding on previous efforts. In particular, the extension to the early days of Landsat acquisitions, enabling the retrieval of a dense Greenland dataset covering nearly 50 years, is of great relevance for exploring factors that are controlling the varying response to climate change for the outlet glaciers in this region and for quantifying their contribution to future sea level rise.

That said, I do think there is some room for improvement of the manuscript, both in terms of presentation as well as substance. What is missing is a clear description of the objectives in the introduction, based on a literature review on the current standing, issues and knowledge gaps in calving front extraction based on machine learning. This gives the reader, not so familiar with the topic, as well as the presented methodological decisions and improvements a better context. Another weak point is that the 'data analysis' does not go any further than a figure showing a rather simple comparison with existing data sets along a flowline of one single glacier. Even though this is clearly written as a methodology paper this is a missed opportunity to showcase a nice data product in my opinion. Perhaps something can be said about general trends in advance/retreat in different regions. Also, I think some sections and descriptions are too brief and need further expansion. Further comments and suggestions for improvement are provided below:

Specific Comments:

Pg 1 – Ln 2: The results uses -> the method uses

Pg 1 – Ln 6: CALFIN provides improvements: briefly describe these improvements

Pg 1 – Ln 7: CALFIN's ability to generalize to SAR imagery is also evaluated: briefly describe the outcome.

Pg 1 – Ln 8: ..deviating by 2.25 px -> deviating by on average 2.25 px

Pg 2 – Ln 4: Previous techniques -> Previous automated techniques

Pg 2 – Ln 3: . . .is a a strong. . . -> is a strong

Pg 2 – Ln 7: Something seems to be missing after this sentence, what has been done already on this topic and what are you going to do/improve in this study? See also above issue raised above.

Pg 2 – Ln 9: Sect 4.1 -> Sect 4

Pg 2 – Ln 9/10: Sect. 5 and Sect. 6 shows as well as discusses the results -> Sect. 5 and Sect. 6 show and discuss the results.

Pg 2 – Ln 12: Sentinel: Sentinel-1 or 2? Not clear from table or text.

Pg 2 – Section 2: This section is too brief and there is no need to add the table if only Landsat data is used in the current work as stated. Aside, it is not clear which Sentinel is meant, e.g. the Sentinel-1 SAR satellite has a repeat cycle of 6/12, not 10/12, Sentinel-2 has 10 days but is optical. Why not use higher resolution 15 m panchromatic band Landsat data?

Pg 2 – Ln 15: The basin selection is based on high drainage volume, based on what source? Also, for robust methodological development it is better to base the selection of study sites on different (fjord/glacier) morphology, scale or front type (e.g. with melange, no melange).

Pg 2 – Ln 20: remove space at beginning.

Pg 3 – Ln 1: This produces -> This results in

Pg 4 – Ln 2: resized: Do you mean crop or actually resize, as the latter would involve changing the resolution?

Pg 4 – Ln 1: ..cloud pixel.. -> how are the cloud pixels identified? Did you include a cloud detection?

Pg 4 – ln 14/16: encoder/decoder: it would be nice to show this in the figure for clarity

Pg 4 – Ln 22: 224 px: wasn't it 256, can you clarify?

Pg 4 – Ln 22: What is the effect of the reduction in input resolution?

Pg 6 – Ln 4: This section is too brief and needs more details on the confidence measure and applied filter criteria.

Pg 6 – Ln 12: Fjord boundary masks: how are these created and based on what source data? Can you expand on this? Also, are they static for the whole time series? I can imagine that ice thinning over several decades affects the ice/ocean/fjord boundary.

Pg 6 – Ln 18: . . .verification each. . . -> verification of each

Pg 7 – Ln 2: error -> the error

Pg 7 – Ln 7: data that is -> data that are

Pg 8 – Ln 2: list tables that print -> show tables with

Pg 8 – Ln 8: CALFIN-VS-L7-only/none: explain what this means

Pg 8 – Ln 11: Antarctic basins: this contradicts Pg 2 - Ln 14 stating that the area of interest is restricted to Greenland

Pg 8 – section 4.3.1: The varying conversion of pixels to distance in this paragraph is confusing, can you clarify this, what is the pixel resolution, how is this calculated, why

[Figure]

Interactive
comment

does it vary?

Pg 9 – Ln 2: generalization capability: please briefly explain what this means.

Pg 9 – section 4.3.3 & 4.3.4: For both intercomparisons the mean pixel distance comparisons is skewed, in the caption of figure 11 it is also mentioned 'undeservedly'. How then can we use this metric to decide which one is better?

Pg 11 – Ln 14: make sure to make this an active link.

Pg 12 – Ln 3-5: Too brief, more discussion needed to explain the loss function.

Pg 12 – Ln 5: Explain what is meant by "over-fitting"

Pg 12 – Ln 12-13: Once. . .processing: sentence incomplete.

Pg 12 – Ln 25: While the methodology is restricted by its preprocessing requirements and inability to handle branching/nonlinear calving fronts: How are the preprocessing requirements different?

Pg 12 – Section 6.2: Some of this existing work description should go to the introduction to show where gaps/shortcomings are and as motivation for the improvements introduced in the current implementation.

Pg 13 – Section 6.3: As mentioned in the general comment, this section is hardly a data analysis and very brief, even the description of the figure. A clear improvement, obvious from the figure, is the much denser and longer temporal coverage, this should be mentioned somewhere.

Pg 13 – Ln 2: validate -> compare

Pg 13 – Ln 7: length change -> I would rather call it "advance and retreat"

Pg 13 – Ln 18/19: To perform . . . the results: this sentence seems incomplete.

Pg 14 – Ln 18: ground truth fronts: None of these fronts are actual ground truth fronts, even when manually delineated (also elsewhere in manuscript).
Pg 15 – Ln 2: Overall, the goal of . . .: this goal was nowhere clearly stated

Figures/Tables

Most figures lack a proper scale bar, this would be very helpful to evaluate the different results. Also, individual lines are sometimes very difficult to distinguish (for example in fig 10). Not sure if this can be improved.

Table 1: As no data other than Landsat is used in the study, I don't see much need for this table. See issue raised previously.

Figure 1: For a nicer figure, updated maps, without gaps, are available at the Greenland Ice Sheet CCI website (see: http://esa-icesheets-greenland-cci.org/)

Figure 2: The legend should provide a range

Figure 3 & 5: No need to add c) in my opinion

Figure 6: It appears that several 'difficult' sections/gaps are connected with a straight line, how does this work (e.g. what gap tresholds are used)?

Figure 6a: I don't see a red coastline mask

Figure 8-12: There seem to be no references in the text to these figures, please add.

Figure 12: caption "Sample" -> Examples

Figure 13: caption "1995-2016 (ESA-CCI), 2005-2017 (MEaSUREs)": check years vs line in image, ESA CCI starts in 1990, MEaSUREs in 2000

---

## Author Comment (AC1) · 18 Jan 2021

**General Comments:**

This manuscript introduces the novel developed Calving Front Machine "CALFIN" for the automated extraction of Greenlandic calving fronts. This is a major contribution to the field as it replaces time-consuming manual delineated fronts by automatically extracted dense glacier front time series. The CALFIN algorithm was validated extensively against test datasets and results from previous studies through a model intercomparison. The scientific community will definitely benefit from this development as an automatically derived calving front position data set of 66 Greenlandic glaciers will be released with this publication.

Despite the impressive results and technical details of this manuscript, I have some concerns about the structure of this paper and the (sometimes) very short explanations. However, after re-structuring some parts of the manuscript and adding additional information as indicated below, this paper will present an important contribution to the field. In my opinion, the abstract should be structured more clearly. For a better understanding, I would recommend to re-order the abstract by using the common schema: 1) Statement of the problem, 2) Research question, 3) Research design, 4) Central results, 5) Brief interpretation of the results, and 6) Outlook/ future use of the data set.

We thank the reviewer for this feedback and have integrated the suggestions into the manuscript. The abstract has now been rewritten according to the standardized schema as follows:

"Sea level contributions from the Greenland Ice Sheet are influenced by the rapid changes in glacial terminus positions. However, the manual delineation of these calving fronts is time consuming, which limits the availability of this data across a wide spatial and temporal range. Automated methods face challenges that include the handling of clouds, illumination differences, sea ice mélange, and Landsat-7 Scanline Corrector Errors. To address these needs, we develop the Calving Front Machine (CALFIN), an automated method for extracting calving fronts from satellite images of marine-terminating glaciers using neural networks. CALFIN's results are often indistinguishable from manually-curated fronts, deviating by on average 86.76 meters ± 1.43 m from the measured front. CALFIN's outputs use Landsat imagery from 1972 to 2019 to generate 22,678 calving front lines across 66 Greenlandic glaciers. This improves on the state of the art in terms of the spatio-temporal coverage and accuracy of its outputs. The current implementation offers a new opportunity to explore sub-seasonal trends on the extent of Greenland's margins, and supplies new constraints for simulations of the evolution of the mass balance of the Greenland Ice Sheet and its contributions to future sea level rise."

P2L4: The paper introduces a new method and provides an inter-comparison with other studies. For readers not familiar with the studies of Zhang et al, Mohajerani et al. and Baumhoer et al. it would be helpful to have a brief state-of-the-art paragraph reviewing existing calving front extraction methods. For example, P2L4 could be extended and give more insights into the studies used in the inter-comparison as well as the studies of Seale et al. 2011 and similar approaches.

These suggestions are appreciated, and we focus on the shortcomings of studies like Seale et al. 2011 to handle Landsat 7 Scanline Corrector Errors, as well as expand upon the state of the art by integrating the Existing Works Sect. 6.2 into the introduction. The edited lines are as follows:

"Existing work by Mohajerani et al. (2019) pioneers the usage of these techniques by applying the Ronneberger et al. (2015) UNet deep neural network towards Jakobshavn, Helheim, Sverdrup, and Kangerlussuaq. It achieves a mean distance error of 96.3 m, but is restricted by the preprocessing requirement of aligning the flow direction to be vertical, and inability to handle branching/non-linear calving fronts. Zhang et al. (2019) evaluates a modified UNet applied to TerraSAR-X data over Jakobshavn, and achieves a mean distance error of 104 m, but is limited in scope. Baumhoer et al. (2019) expands the application of the UNet to Sentinel 1 imagery of Antarctica, extracting full coastline delineations and achieving a mean distance error of 108 m. Ultimately, these case studies provide the groundwork for the automatic, accurate, large scale, longtime-series, high temporal resolution, and potentially multi-sensor extraction of glacial terminus positions."

P2L11: In my opinion this section is incomplete. Please mention all potential data sources in Table 1 (add Sentinel-2, Envisat, ERS, Radarsat) and justify why they are not suitable. Another option would be to just focus on Landsat data and remove the incomplete Table 1. Figure 1 is really great so I would try to put the focus on it and highlight the incredible amount of processed data and outline the advantages, data amount, and characteristics of Landsat.

Thank you for these comments - Table 1 has been removed in favor of elaborating on the advantages/characteristics of the data sources evaluated in the study, which now covers Sentinel 1A/B as well.

P2L17: The methodology section could give a short overview of the entire workflow from pre-processing to the final extracted calving front by showing a flow chart. This would guide the reader through the methodology part and link the numerous subchapters of section 3. Besides, in my opinion, the training of the network explained in P12L2 should be part of the methodology and not subject to the discussion.

These are good points, and a methodology flowchart has been added to the beginning of Sect. 3 (see Fig. R1 below). Additionally, the network training discussion subsection Sect. 6.1 has been integrated into the methodology as Sect 3.2p4.

[Figure]

**Figure R1. CALFIN Processing Flowchart**

**Specific Comments:**

P5 Figure 5c: How does the filtering of unconfident predictions work? Please describe this in the methodology section.

The filtering of unconfident predictions is performed by measuring the certainty of each pixel's classification in a 5 pixel wide buffer around the calving front. Predictions with a mean certainty exceeding an empirically chosen threshold will be filtered from the results. The following explanation of the method is now given at the end of Sect 3.3p4:

"Since the neural network assigns each pixel a value between 0 and 1 based on its perceived class, any deviation from these two values can used as a measure of uncertainty. The filtering method averages the deviation of the ice/ocean classification mask in a 5 pixel wide buffer around the calving front, and discards any fronts whose mean deviation exceeds an empirically chosen threshold of 0.125."

P6L1: Please outline the calving front re-processing in more detail. Does the reprocessing allow a higher spatial accuracy when re-processing a part of the image?

Yes, the reprocessing allows for higher spatial accuracy when re-processing the image. The re-processing step is now more clearly shown in the Fig. R1 flowchart and described at the beginning of Sect 3.3p4: "Once each front is located, its bounding box is used to extract a higher resolution subset from the original image, and reprocessed. This innovation allows for increased spatial accuracy when processing multiple fronts in large basins."

P6L16: How much smoothing of the extracted coastline is allowed and can this also decrease accuracy?

The smoothed coastline is allowed to vary by no more than 1 pixel from the raw extracted coastline, as seen in Fig. R2. Since the variations are on the sub-pixel scale, the error introduced is no more than the uncertainty of the base resolution, and well within the neural network uncertainty. The following clarification has been added to the end of the line: ", deviating no more than 1 pixel from the raw extracted coastline.". Fig. R2 has also been added to the Supplement as Fig. S2.

[Figure]

Figure R2. Smoothed (Orange) Versus Raw Coastline (Blue)

P8L1: How did you handle the issue that your network was trained for 3-channel RGB imagery but tested on 1-channel SAR data?

This is question is appreciated, as it highlights the manuscript's shortcomings in describing the SAR preprocessing pipeline. A paragraph has been added in Sect. 2, Data Source and Scope, describing the usage of the Sentinel 1A/B Antarctic SAR HH band to measure backscatter intensity, which is then treated the same as a Landsat 1-channel NIR band and preprocessed into the final 3-channel false color RGB imagery.

The flowchart in Fig. R1 also helps clarify the input preprocessing steps needed to derive a 3-channel false color RGB image from 1-channel input rasters (now Fig. 3 in the manuscript).

P8L18: What are the characteristics of those outlier glaciers and how many glaciers are defined as "outlier"?

Glaciers with ice tongues such as Kong Oscar can result in large disagreements between the predicted front and the manually delineated fronts. Kong Oscar is the only glacier in the CALFIN Validation Set that contains such extensive ice tongues.

Since the "outlier" in this line refers only to the statistical outlying measurements, and no glaciers are excluded from the error metric calculations, the clause "When excluding outliers such as Kong Oscar, " has been removed to reduce confusion.

P11L15: The information of this section could also be shifted to methodology. Then rename Chapter 5 to "CALFIN Dataset".

Thank you for this suggestion - this change has been integrated, and Sect. 5.2 has been removed.

P10L4: But also mention the mean distance which is comparable here.

These lines have been rewritten to include the mean distance error as follows:

"When comparing the mean distance error with the Baumhoer et al. (2019) equivalent Area over Front (A/F) error, the Baumhoer et al. (2019) neural network (B-NN) outperforms CALFIN-NN (330.63 m vs 108 m). Note that the easily detected static coastlines are masked out, raising the relative error, and negatively impacting CALFIN-NN's performance on this metric."

P10 Figure 11: How did you consider the fact that ice shelves are much bigger than glaciers? For example, in Figure 11 you show the Shackleton ice shelf. It is approx. 200 km wide and if you resample that to 224x224 pixels, one pixel for your validation would be 892 m compared to 40 m pixels in the original study by Baumhoer et al. 2019. How did this influence the validation accuracy? For Zhang et al. you show that the use of higher resolution of TerraSAR-X data does not improve the mean distance accuracy (Figure 10).

Errors in large ice shelves are the primary contributor to CALFIN's large mean distance error values. For Shackleton ice shelf, the highly accurate detection prevents it from contributing excessive amounts of error, though indeed variations of even 1 pixel would cause significant error. The following graphs (Fig. R3-R5) shows a histogram that plots the distance between closest pixels in the predicted and manually delineated 3-pixel wide calving front masks. Shackleton's mean distance of 287.48 meters (Fig. R3) for a single validation image is better than the overall average (330 meters) when compared to other large domains like Voyeykov (Fig. R4) and Land (Fig. R5).

[Figure]

Figure R3. Shackleton Pixelwise Mean Distance Error Histogram

[Figure]

**Figure R4. Voyeykov Pixelwise Mean Distance Error Histogram**

[Figure]

**Figure R5. Land Pixelwise Mean Distance Error Histogram**

For Zhang et al., the higher resolution inputs are resized to a lower resolution to fit into the 224x224 neural network input shape, and thus provides no improvements. A neural network with a larger input size would benefit from higher resolution imagery.

P13 Figure 13: Can you explain why the PROMICE data set (2008/2009 and 2010/2011) shows twice a very different front position compared to the CALFIN data set?

PROMICE (Anderson et al., 2019) does not provide dates for its delineations, instead stating that they are observed at the "end-of-melt season". August 15th was chosen as the apparent date of these measurements, and it generally corresponds to the other measurements, but it is not a reliable indicator of the calving front at sub-annual timescales, and is only provided for context.

P13L13: The model inter-comparison is only discussed for the study of Mohajerani et al. but validations were also done against the data sets of Zhang et al. and Baumhoer et al., hence those results should also be discussed.

This is a valuable suggestion, and should be investigated in a follow up study, but is unfortunately out of the scope of this study due to the computational and logistical challenges of retraining the original networks used in Zhang et al. and Baumhoer et al. with the CALFIN training set, and the necessary involvement of the original authors in such an in-depth intercomparison.

---

## Author Comment (AC2) · 18 Jan 2021

**General Comments:**

General Comment Cheng et al. present an automated method for delineating glacier calving fronts – named Calving Front Machine (CALFIN) - based on a deep learning approach, accompanied by a new dataset of Greenland glacier termini. The principal input data are Landsat optical images acquired since 1972. The methodology builds on previous work by Mohajerani et al., Zhang et al., and Baumhoer et al. and uses computing systems, named neural networks, that learn patterns in training data, in order to identify similar patterns (such as glacier termini) in new data. The authors detail the various steps of the processing chain and produce a set of shapefiles, which are evaluated and intercompared with both internal and external (manually) retrieved calving front datasets using different quality metrics. The main outcome is an extensive dataset covering 66 outlet glaciers around Greenland with in total 22,679 individual calving fronts encompassing the period 1972-2019. The method and new data set reportedly exceeds the accuracy of previous work and approaches human levels of accuracy in delineating glacier termini, the key takeaway being the maturation of neural networks for automated calving front detection.

Automated calving front extraction is a long sought after goal, that recently gained new attention thanks to advances in modern computing technology and increasing availability of satellite EO data. The use of deep learning/neural networks – the subject of this paper - to achieve this is very promising indeed. This paper by Cheng et al. is a welcome addition to existing literature on this topic as is the associated dataset for the community, expanding on previous efforts. In particular, the extension to the early days of Landsat acquisitions, enabling the retrieval of a dense Greenland dataset covering nearly 50 years, is of great relevance for exploring factors that are controlling the varying response to climate change for the outlet glaciers in this region and for quantifying their contribution to future sea level rise.

That said, I do think there is some room for improvement of the manuscript, both in terms of presentation as well as substance. What is missing is a clear description of the objectives in the introduction, based on a literature review on the current standing, issues and knowledge gaps in calving front extraction based on machine learning. This gives the reader, not so familiar with the topic, as well as the presented methodological decisions and improvements a better context.

We thank the reviewer for their time, comments, and suggestions, which have been integrated into the manuscript. A clear description of the objective has been added to the introduction and abstract. This is based on issues and knowledge gaps covered in the added literature review, which repurposes existing sections to provide better methodological context. Additional references have been added throughout the introduction, and a new paragraph has been integrated as follows: "Existing work by Mohajerani et al. (2019) pioneers the usage of these techniques by applying the Ronneberger et al. (2015) UNet deep neural network towards Jakobshavn, Helheim, Sverdrup, and Kangerlussuaq. It achieves a mean distance error of 96.3 m, but is restricted by the preprocessing requirement of aligning the flow direction to be vertical, and inability to handle branching/non-linear calving fronts. Zhang et al. (2019) evaluates

a modified UNet applied to TerraSAR-X data over Jakobshavn, and achieves a mean distance error of 104 m, but is limited in scope. Baumhoer et al. (2019) expands the application of the UNet to Sentinel 1 imagery of Antarctica, extracting full coastline delineations and achieving a mean distance error of 108 m. Ultimately, these case studies provide the groundwork for the automatic, accurate, large scale, longtime-series, high temporal resolution, and potentially multi-sensor extraction of glacial terminus positions."

Another weak point is that the 'data analysis' does not go any further than a figure showing a rather simple comparison with existing data sets along a flowline of one single glacier. Even though this is clearly written as a methodology paper this is a missed opportunity to showcase a nice data product in my opinion. Perhaps something can be said about general trends in advance/retreat in different regions. Also, I think some sections and descriptions are too brief and need further expansion. Further comments and suggestions for improvement are provided below:

Several sections have been expanded based on provided feedback. Additionally, the data analysis has been expanded, with a new figure showing the regional trends for NW, CW, CE, SW, and SE Greenland, along with 9 additional glacial flowline graphs:

"

[Figure]

Figure 14. Regional Terminus Advance and Retreat Over Time. (a-f) Regional delineations (left) and terminus position graphs (right)for Greenland (a) and the northwestern (b), central western (c), central eastern (d), southeastern (e), and southwestern (f) regions. Note that the total Greenland mean advance and retreat is unadjusted, and dominated by the trend lines of numerous smaller glaciers in CW and NW Greenland. Note that branches in the 66 studied basins are independently counted, for a total of 87 glaciers.

Additionally, Fig. 14 shows the regional mean advance and retreat change, alongside the mean for the entirety of Greenland covered by the CALFIN dataset. Contributions from NW Greenland influence the overall trend the most, due to the presence of many small glaciers/branches in the regions. Note that the mean for Greenland also includes contributions from Petermann, which is visible in the summers of 2010 and 2012. Shared

regional trends are visible across NW and CW Greenland, which both show relative stability before 2000, followed by steady retreat up until 2017-2018. CE and SE Greenland also share similar but less pronounced retreat, showing accelerating retreat beginning around 1995. These regional trends are less visible in SW Greenland, which is dominated by Narsap Sermia's retreat from 2010-2013. Overall, these regional trends generally agree with studies such as Wood et al. (2021) and King et al. (2020), helping further validate the CALFIN method and data."

Specific Comments:

Pg 1 – Ln 2: The results uses -> the method uses

Done.

Pg 1 – Ln 6: CALFIN provides improvements: briefly describe these improvements

Among existing works, CALFIN improves on the spatial accuracy, is applied towards a large selection of glacial basins, and provides the outputs for scientific usage. "…improvements on the current state of the art." is now described as "…improves on the state of the art in terms of the spatio-temporal coverage and accuracy of its outputs."

Pg 1 – Ln 7: CALFIN's ability to generalize to SAR imagery is also evaluated: briefly describe the outcome.

CALFIN is able to process SAR imagery with similar levels of accuracy when compared to its performance on Landsat image, and is competitive with existing studies. "CALFIN's ability to generalize to SAR imagery" has been moved from the abstract and expanded upon in Sect 2. (see the response to Pg 2 – Ln 12).

Pg 1 – Ln 8: ..deviating by 2.25 px -> deviating by on average 2.25 px

Done.

Pg 2 – Ln 4: Previous techniques -> Previous automated techniques

Done.

Pg 2 – Ln 3: . . .is a a strong. . . -> is a strong

Fixed.

Pg 2 – Ln 7: Something seems to be missing after this sentence, what has been done already on this topic and what are you going to do/improve in this study? See also above issue raised above.

Thank you for raising these points - the section has been expanded upon, and now includes a literature review of existing work and a statement of goals. The added text is as follows:

"Existing work by Mohajerani et al. (2019) pioneers the usage of these techniques by applying the Ronneberger et al. (2015) UNet deep neural network for towards Jakobshavn, Helheim, Sverdrup, and Kangerlussuaq. It achieves a mean distance error of 96.3 m, but is restricted by the preprocessing requirement of aligning the flow direction to be vertical, and inability to handle branching/non-linear calving fronts. Zhang10et al.

(2019) evaluates a modified UNet applied to TerraSAR-X data over Jakobshavn, and achieves a mean distance error of104 m, but is limited in scope. Baumhoer et al. (2019) expands the application of the UNet to Sentinel 1 imagery of Antarctica, extracting full coastline delineations and achieving a mean distance error of 108 m. Ultimately, these case studies provide the groundwork for the automatic, accurate, large scale, long time-series, high temporal resolution, and potentially multi-sensor extraction of glacial terminus positions. This study seeks to assess the feasibility of achieving robust automatic extraction for a15selection of Greenland's glaciers, and to provide the resulting dataset for use by the wider community. Additionally, this study seeks to assess improvements to the neural network design and post-processing methods."

Pg 2 – Ln 9: Sect 4.1 -> Sect 4

Fixed.

Pg 2 – Ln 9/10: Sect. 5 and Sect. 6 shows as well as discusses the results -> Sect. 5 and Sect. 6 show and discuss the results.

Done.

Pg 2 – Ln 12: Sentinel: Sentinel-1 or 2? Not clear from table or text.

We use Sentinel 1 - this is now addressed by a new paragraph at the end of Sect. 2, describing the addition of Sentinel 1A/B Antarctic SAR data for the sole purposes of training and validating the CALFIN methodology. We have added the following new paragraph to this Section:

"For the training and validation of the CALFIN methodology, Sentinel 1A/B SAR images are added to enforce the applicability of the method to other sensor types and domains. The area of interest for the training and validation of the methodology thus includes Antarctic SAR data in addition to the Greenlandic Landsat optical data (see Sect. and Fig. S4). The product used is the Extra Wide Swath, Ground Range Multi-Look Detected, 40 meter resolution HH polarization band. The other data products and polarization bands are not used since the HH backscatter intensity provides sufficient information for the data processing methodology to succeed. A characteristic of Sentinel 1A/B - and SAR data in general - is the presence of speckle noise, which is addressed by the methodology described in the following section."

Pg 2 – Section 2: This section is too brief and there is no need to add the table if only Landsat data is used in the current work as stated. Aside, it is not clear which Sentinel is meant, e.g. the Sentinel-1 SAR satellite has a repeat cycle of 6/12, not 10/12, Sentinel-2 has 10 days but is optical. Why not use higher resolution 15 m panchromatic band Landsat data?

Thank you for raising these points – the first is addressed by the revisions to Sect. 2, which describes the use of Landsat data for dataset production, and both Landsat as well as Sentinel 1A/B data for training and validation.

The 15-meter resolution panchromatic band is not used due to resolution bottlenecks in the data processing methodology. In other words, the increase in resolution did not provide significant increases in accuracy, as it would be downscaled to the same

resolution as the 30 meter inputs to fit the small neural network input size. This clarification has been added to the end of the first paragraph in Sect. 2.

Pg 2 – Ln 15: The basin selection is based on high drainage volume, based on what source? Also, for robust methodological development it is better to base the selection of study sites on different (fjord/glacier) morphology, scale or front type (e.g. with melange, no melange).

The selection metric is based off the basin area/velocities from Nagler et al., 2015. The basins are indeed also selected for robust methodological development, and the 10 areas of interest as well as any nearby basins were selected to contain unique features like ice tongues, branches, and various mélange types. The line now states this explicitly as "The basins are selected for their high drainage volume, wide spatial distribution, and diverse morphological features."

Pg 2 – Ln 20: remove space at beginning.

Fixed.

Pg 3 – Ln 1: This produces -> This results in

Done.

Pg 4 – Ln 2: resized: Do you mean crop or actually resize, as the latter would involve changing the resolution?

The subsets are resized, and the resolution is indeed changed. This loss of resolution is addressed by the reprocessing step, where the subset is recropped at the original resolution and resized again, to allow for maximum resolution within the constraints of the neural network input size.

Pg 4 – Ln 1: ..cloud pixel.. -> how are the cloud pixels identified? Did you include a cloud detection?

The cloud pixels are identified using the Landsat QA band, which assigns each pixel a value based on its detected cloud coverage. The line has been clarified as "…cloud pixels detected in the Landsat QA band,". We rely on the provided cloud masks given by Landsat to do additional filtering per subset, as the scene cloud cover filtering only filters raster based on whole scene cloud coverage.

Pg 4 – ln 14/16: encoder/decoder: it would be nice to show this in the figure for clarity

Done.

Pg 4 – Ln 22: 224 px: wasn't it 256, can you clarify?

The 256px subsets are split into 9 224 px overlapping windows. The Sect. 3, Methodology flowchart (Fig. 3) and Sect. 3.2p4 now clarifies this apparent discrepancy.

Pg 4 – Ln 22: What is the effect of the reduction in input resolution?

This is a good question, as the reduction of input resolution allows for greater complexity, faster training, and higher practical accuracy of the model, but limits the maximum theoretical spatial accuracy of the network. We use other methods (such as

overlapping subsets) to extract higher accuracy predictions from the lower input resolution model.

These considerations have been clarified, and the line has been rephrased to state how reducing the input size results indirectly in increased accuracy, from "To facilitate faster training and performance, the input size is reduced from 512 px to 224 px" to "The input size is reduced from 512 px to 224 px to facilitate better computational performance, allowing for additional training and thus higher accuracy".

Pg 6 – Ln 4: This section is too brief and needs more details on the confidence measure and applied filter criteria.

This is a fair point - the section has been expanded, and surrounding sections have been rearranged to better support the new narrative. The added material is as follows:

"Once each front is located, its bounding box is used to extract a higher resolution subset from the original image, and reprocessed. This innovation allows for increased spatial accuracy when processing multiple fronts in large basins. After reprocessing, the nature of CALFIN-NN's dual outputs as a confidence measure is exploited to filter and discard uncertain detections. Since the neural network assigns each pixel a value between 0 and 1 based on its perceived class, any deviation from these two values can be used as a measure of uncertainty. The filtering method averages the deviation of the ice/ocean classification masking a 5 pixel wide buffer around the calving front, and discards any fronts whose mean deviation exceeds an empirically chosen threshold of 0.125."

Pg 6 – Ln 12: Fjord boundary masks: how are these created and based on what source data? Can you expand on this? Also, are they static for the whole time series? I can imagine that ice thinning over several decades affects the ice/ocean/fjord boundary.

Thank you for these questions and comments - the masks are static and manually created using the image subsets and BedMachine V3 for reference. They are static and averaged across the whole time series – while there are indeed minor changes in the coastline over this time, they do not affect the accuracy of the calving front delineation within the fjord.

This has been clarified as "Static masks of the average fjord boundaries are first created for each basin using the image subsets and BedMachine V3 for reference"

Pg 6 – Ln 18: . . .verification each. . . -> verification of each

Fixed.

Pg 7 – Ln 2: error -> the error

Fixed.

Pg 7 – Ln 7: data that is -> data that are

Fixed.

Pg 8 – Ln 2: list tables that print -> show tables with

Done.

Pg 8 – Ln 8: CALFIN-VS-L7-only/none: explain what this means

A new sentence has been added to this section, which now defines CALFIN-VS-L7-only/ CALFIN-VS-L7-none: "To evaluate performance on Landsat 7 Scanline Corrector Errors, the validation subset CALFIN-VS-L7-only isolates images with L7SCEs, and the CALFIN-VS-L7-none excludes images with L7SCEs."

Pg 8 – Ln 11: Antarctic basins: this contradicts Pg 2 - Ln 14 stating that the area of interest is restricted to Greenland

This observation is appreciated - the response to Pg 2 – Ln 12 addresses this by adding a new paragraph at the end of Sect. 2, Data Source and Scope, describing the addition of Antarctic SAR data for the sole purpose of training and validating the CALFIN methodology.

Pg 8 – section 4.3.1: The varying conversion of pixels to distance in this paragraph is confusing, can you clarify this, what is the pixel resolution, how is this calculated, why does it vary?

The pixel conversion varies due to 2 effects: images are reprocessed at lower sizes due to detection failures (see Fig. 5c), and pixel error increases as resolution decreases (see Sect. 4.1). Since the pixel-to-meter rate is depends on the scaling factor of each subset, the distribution of rates changes as Landsat 7 images are added/removed.

The methodology flowchart and the elaboration of the filtering/reprocessing step should make this interaction of effects more understandable.

Additionally, the addition of scales to the subsets should aid in communicating the different pixel to meter conversion ratios per subset.

Furthermore, the pixel error metrics have been removed from the paragraph to reduce confusion and to not detract from the more intuitive meter error metrics.

Pg 9 – Ln 2: generalization capability: please briefly explain what this means.

In this context, generalization capability is the ability of a neural network to accurately make new predictions on data it has not been trained on before.

The line "This demonstrates the generalization capability of CALFIN-NN" has been clarified as "This demonstrates CALFIN-NN's ability to accurately process new data".

Pg 9 – section 4.3.3 & 4.3.4: For both intercomparisons the mean pixel distance comparisons is skewed, in the caption of figure 11 it is also mentioned 'undeservedly'. How then can we use this metric to decide which one is better?

This is a good question - the mean pixel distance metric can be used to decide which network is better only when comparing neural networks of the same input size. Indeed, the metric is not useful when comparing networks of different input sizes, since it favors smaller input sizes.

We still provide the metric for comparison to provide additional context when comparing CALFIN with existing studies, as these studies have done the same.

Pg 11 – Ln 14: make sure to make this an active link.

Fixed and verified.

Pg 12 – Ln 3-5: Too brief, more discussion needed to explain the loss function.

Thanks for this noting this shortcoming in the manuscript - a more detailed explanation and relevant equations have been added as follows:

"To increase accuracy, a custom loss function optimizes the binary cross entropy and Intersection-over-Union (see Eq. 1, Sect.4.1). This penalizes mismatches between calving front pixels in the predicted ($\mathbf{I_{cf}}$) and measured ($\mathbf{\hat{I}_{cf}}$) image masks. Similarly mismatched ice/ocean pixels in the predicted ($\mathbf{I_{io}}$) and measured ($\mathbf{\hat{I}_{io}}$) image masks are less heavily weighted by an empirically chosen factor of $\alpha = 1/25$, as seen in the final loss function $\mathbf{L}$ in Eq. 2."

$$BCE\_IoU(\mathbf{I}, \mathbf{\hat{I}}) = -\mathbf{I} \cdot \log(\mathbf{\hat{I}}) - (1 - \mathbf{I}) \cdot \log(1 - \mathbf{\hat{I}}) - \log\left(\frac{\mathbf{I} \cap \mathbf{\hat{I}}}{\mathbf{I} \cup \mathbf{\hat{I}}}\right) \tag{1}$$

$$\mathcal{L}(\mathbf{I_{cf}}, \mathbf{\hat{I}_{cf}}, \mathbf{I_{io}}, \mathbf{\hat{I}_{io}}) = \alpha \cdot BCE\_IoU(\mathbf{I_{io}}, \mathbf{I_{io}}) + (1 - \alpha) \cdot BCE\_IoU(\mathbf{I_{cf}}, \mathbf{\hat{I}_{cf}}) \tag{2}$$

Pg 12 – Ln 5: Explain what is meant by "over-fitting"

In this context, "over-fitting" means that the model has been trained too heavily on a small dataset, and has only effectively memorized it instead of learning more general features of the observed data. This prevents it from accurately making predictions on new data, as it has "over-fit" the training data.

These lines have been rephrased to be clearer, from "To prevent over-fitting the neural network" to "In order to train the neural network", and from "Another measure to prevent over-fitting involves data augmentation" to "Data augmentation is used during training to increase the accuracy of the network when processing new data".

The only other instance of "over-fitting" on Pg 15, Sect 6.4. is elaborated as "over-fitting, or memorizing,".

Pg 12 – Ln 12-13: Once. . .processing: sentence incomplete.

Thanks for catching this error. The line has been fixed and rephrased from "Once trained, an NVIDIAGTX 1080 with 6GB VRAM for off-line data processing" to "Once trained, an NVIDIA GTX1060 with 6GB VRAM is used for the off-line data processing of the 20188 GeoTIFF subsets". The phrase "of the 20188 GeoTIFF subsets" has been moved from a subsequent line to clarify what data is being processed off-line.

Pg 12 – Ln 25: While the methodology is restricted by its preprocessing requirements and inability to handle branching/nonlinear calving fronts: How are the preprocessing requirements different?

The primary difference in preprocessing requirements is the necessary alignment of the flow direction to be vertical. This line has been elaborated as "the preprocessing requirement of aligning the flow direction to be vertical".

Pg 12 – Section 6.2: Some of this existing work description should go to the introduction to show where gaps/shortcomings are and as motivation for the improvements introduced in the current implementation.

Thank you for this suggestion - Sect. 6.2 has been integrated into the introduction, along with descriptions of the gaps/shortcomings of each approach that form the motivation for the study. The end of the first paragraph of the introduction now reads, "Existing work by Mohajerani et al. (2019) pioneers the usage of these techniques by applying the Ronneberger et al. (2015) UNet deep neural network for towards Jakobshavn, Helheim, Sverdrup, and Kangerlussuaq. It achieves a mean distance error of 96.3 m, but is restricted by the preprocessing requirement of aligning the flow direction to be vertical, and inability to handle branching/non-linear calving fronts. Zhang et al. (2019) evaluates a modified UNet applied to TerraSAR-10X data over Jakobshavn, and achieves a mean distance error of 104 m, but is limited in scope. Baumhoer et al. (2019) expands the application of the UNet to Sentinel 1 imagery of Antarctica, extracting full coastline delineations and achieving a mean distance error of 108 m. Ultimately, these case studies provide the groundwork for the automatic, accurate, large scale, longtime-series, high temporal resolution, and potentially multi-sensor extraction of glacial terminus positions. This study seeks to assess the feasibility of achieving robust automatic extraction for a selection of Greenland's glaciers, and to provide the resulting dataset for use by the wider community. Additionally, this study seeks to assess improvements to the neural network design and post-processing methods."

Pg 13 – Section 6.3: As mentioned in the general comment, this section is hardly a data analysis and very brief, even the description of the figure. A clear improvement, obvious from the figure, is the much denser and longer temporal coverage, this should be mentioned somewhere.

Thank you for pointing out this weakness in the original manuscript. The figure description has been expanded with the following details: "Note the seasonal variations shown by the solid lines, and the dotted lines from 1972-1985 that indicate a lack of such seasonal observations. Also note that the vertical axis scaling is applied differently for each graph to highlight seasonal trends." Text that highlights the denser and longer temporal coverage has been added throughout the section. Furthermore, the original Fig. 12 (now Fig. 13) has been expanded to include additional flowlines:

[Figure]

**Figure R1. Updated Terminus Advance and Retreat Over Time**

See also the response to the general comment for additional added content that adds to the data analysis.

Pg 13 – Ln 2: validate -> compare

Done.

Pg 13 – Ln 7: length change -> I would rather call it "advance and retreat"

Done.

Pg 13 – Ln 18/19: To perform . . . the results: this sentence seems incomplete.

Thank you for noticing this – the sentence has been rephrased for clarity, from "To perform this task, the M-NN is retrained using CALFIN training data, process validation data, and compare the results" to "This task involves retraining the M-NN on CALFIN training data, and comparing its performance against CALFIN-NN using a shared validation set".

Pg 14 – Ln 18: ground truth fronts: None of these fronts are actual ground truth fronts, even when manually delineated (also elsewhere in manuscript).

This is a good point, and has been corrected from "ground truth" to "manually delineated" throughout the manuscript.

Pg 15 – Ln 2: Overall, the goal of . . .: this goal was nowhere clearly stated

This observation is appreciated, and the introduction has been edited to include this goal, which is stated as, "This study seeks to assess the feasibility of achieving robust automatic extraction for a selection of Greenland's glaciers, and to provide the resulting

dataset for use by the wider community. Additionally, this study seeks to assess improvements to the neural network design and post-processing methods."

**Figures/Tables:**

Most figures lack a proper scale bar, this would be very helpful to evaluate the different results. Also, individual lines are sometimes very difficult to distinguish (for example in fig 10). Not sure if this can be improved.

Thank you for this suggestion - scale bars have been added in Figs. 9-13, and high contrast colorblind-friendly line colors have been added for Figs. 6-12.

Table 1: As no data other than Landsat is used in the study, I don't see much need for this table. See issue raised previously.

Table 1 has been removed.

Figure 1: For a nicer figure, updated maps, without gaps, are available at the Greenland Ice Sheet CCI website (see: http://esa-icesheets-greenland-cci.org/)

Thanks for this suggestion, Fig. 1 has been updated to utilize an updated gapless velocity map.

Figure 2: The legend should provide a range

Fig. 2 key has been updated to show the full range of the data.

Figure 3 & 5: No need to add c) in my opinion

This is a fair suggestion that highlights the lack of importance placed on the filtering step in the manuscript. To address this concern, Fig. 3 & 5 (now 4 & 6) have added a visualization of the filtering under (c), as shown in the new flowchart (now Fig. 3).

Figure 6: It appears that several 'difficult' sections/gaps are connected with a straight line, how does this work (e.g. what gap tresholds are used)?

This is a valuable question that highlights the manuscript's insufficient explanation of this algorithm. Gaps are given negative exponential distance-based weights, so that they add a penalty to the maximum path, but can be used if they connect two long paths in the final Minimum Spanning Tree. An explanation of this behavior has been added to the end of Sect. 3.3.1: "Such gaps are given weights based on the negative exponential distances between nodes, which allows for connections if the paths connected are significantly longer than the gap itself."

Figure 6a: I don't see a red coastline mask

Fig. 6 (now Fig. 7) has been updated to use a high contrast colorblind-friendly color scheme, and the red coastline mask has been enhanced to make it more visible.

Figure 8-12: There seem to be no references in the text to these figures, please add.

Thank you for noting this, references to these figures (now Fig. 9-13) have been added in the text.

Figure 12: caption "Sample" -> Examples

Done.

Figure 13: caption "1995-2016 (ESA-CCI), 2005-2017 (MEaSUREs)": check years vs line in image, ESA CCI starts in 1990, MEaSUREs in 2000

Fixed.

---

## Author Comment (AC3) · 18 Jan 2021

**Calving Front Machine (CALFIN): Glacial Termini Dataset and Automated Deep Learning Extraction Method for Greenland, 1972-2019**

Daniel Cheng[1], Wayne Hayes[1], Eric Larour[2], Yara Mohajerani[1,3], Michael Wood[2], Isabella Velicogna[1,2], and Eric Rignot[1,2]

[1]University of California at Irvine, Irvine CA, USA
[2]Jet Propulsion Laboratory, California Institute of Technology, Pasadena CA, USA
[3]University of Washington, eScience Institute and Department of Civil and Environmental Engineering, Seattle, WA, 98195, USA

**Correspondence:** Daniel Cheng (dlcheng@uci.edu)

**Abstract.**  Sea level contributions from the Greenland Ice Sheet are influenced by the rapid changes in glacial terminus positions. However, the manual delineation of these calving fronts is time consuming, which limits the availability of this data across a wide spatial and temporal range. Automated methods face challenges that include the handling of clouds, illumination differences, sea ice mélange, and Landsat-7 Scanline Corrector Errors. To address these needs, we develop the Calving Front Machine (CALFIN), an automated method, for extracting calving fronts from satellite images of marine-terminating glaciers  using neural networks. The results are often indistinguishable from manually-curated fronts, deviating by on average 86.76 meters ± 1.43 m from the measured front. Landsat imagery from 1972 to 2019  is used to generate 22,678 calving front lines across 66 Greenlandic glaciers. ~~Our method uses deep learning, and builds on existing work by Mohajerani et al., Zhang et al., and Baumhoer et al. Additional post-processing techniques allow our method to achieve accurate segmentation of imagery into Shapefile outputs. This method is uniquely robust to the impact of clouds, illumination differences, ice mélange, and Landsat-7 Scan Line Corrector errors. CALFIN provides improvements on the current. We show this by performing a model inter-comparison and evaluate performance against existing methodologies. We also evaluate 
[revised manuscript text omitted]

**5.1 CALFIN-NN Implementation**

We release an implementation of CALFIN-NN, available at , which includes the parameters and architecture we develop throughout this study. It is our intention that any innovations as described in Sect. 3.2 can be applied to other networks and investigations. The implementation is written in Python 3 using the Keras & Tensorflow libraries. Note that access to the network parameters are also hosted as part of the associated DataDryad dataset linked above (Cheng et al., 2020). For additional insight into the network training and processing requirements, see the following discussion in Sect. **??**.

**6 Discussion**

**5.1 Training Insights**

Throughout the course of the study, we develop several innovations to improve the performance of CALFIN-NN. To increase accuracy, we utilize a special loss function that heavily favors correct calving front predictions. To prevent over-fitting our 
[revised manuscript text omitted]

---

## Author Response (AR2)

**Editor Comment #2**

Dear Daniel Cheng and co-authors,

Your manuscript tc-2020-231 "Calving Front Machine (CALFIN): Glacial Termini Dataset and Automated Deep Learning Extraction Method for Greenland, 1972–2019" has again been reviewed by both reviewers both suggested your revised manuscript could be accepted subject to minor revisions. Moreover, they both added minor comments that should be incorporated.

Based on this assessment, I recommend that your paper can be "Published subject to minor revisions" once the suggested changes are implemented and clarified my means of an author response letter.

Additionally I would like to ask you to update Figure 13 and 14. In its current setup it is impossible to read the labels, color bar legend, etc. Please update the figure in order to make it also readable without a magnifier.

Best regards,

Stef Lhermitte

We thank the editor again for the continued contributions to this manuscript throughout the review process. In this revision, Figures 13 and 14 have been updated to enhance readability at standard magnification. This includes the enlarging of the labels, scale bars, and legends for all subplots. The positioning of the legends and the simplified, high-contrast text also enhance the figures. Furthermore, as per suggestions from Anonymous Referee #1 Report #2, the individual subplots for Figs. 13-14 are now included in the Supplement as Figs. S13-S28. Figure 14 now also uses a shared vertical scale, ranging from -1 to 6.5 km. The updated figures are provided below for reference.

**Relative Advance and Retreat, 1972-2019**

[Figure]

**Figure R1. Selected Terminus Advance and Retreat Graph.**

[Figure]

**Figure R2. Regional Mean Terminus Advance and Retreat Graph**

**Anonymous Referee #1 Report #2**

**General Comments:**

The revised version of the manuscript from Cheng et al. improved significantly. The authors addressed all concerns appropriately and were able to increase the quality of the paper in terms of visualization and written explanation. The revised abstract addresses a broader audience and the restructuring of the chapters allows for a better understanding of the work performed. Especially, the added Flowchart (Figure 3) as well as Figures 13 & 14 highlight the developed methodology and amount of processed data. Finally, the revised manuscript is in line with the great methodological approach of this study.

Only some minor remarks remain which are indicated below.

We thank the reviewer again for the feedback and contributions to the improved manuscript, and address the specific remarks as follows.

**Specific Comments:**

P1L6: change "average 86.76 meters ± 1.43 m" to "average 86.76± 1.43 meters"

Fixed.

P2L31: Also mention TerraSAR-X validation data used for the comparison with the study from Zhang et al.

This comment is appreciated, and lines have been added to clarify the usage of TerraSAR-X data in this study.

The first line of Sect. 2p2 now reads: "For the training and validation of the CALFIN methodology, TerraSAR-X and Sentinel 1A/B SAR images are added to enforce the applicability of the method across different sensors and domains."

The third line of Sect. 2p2 has been added as: "The TerraSAR-X product used is the StripMap 3 meter resolution HH polarization band."

P6L5: Why were rotations up to 12° chosen?

We determined through empirical testing that excessive image rotations resulted in sub-optimal performance – this is due to two factors, which are the introduction of blank/invalid pixels along the corners of the newly rotated image, along with scaling effects when performing a rotation and center crop to only include non-blank/valid pixels.

P8L9: can be used

We have clarified this sentence to read as follows: "This path not only corresponds with the coastline edge, but also out-performs outputs from other contour finding algorithms by eliminating noise, errors, and gaps inherited from previous steps."

Figure 13: This Figure represents a great improvement and shows the amount of processed data. Unfortunately, it is difficult to read the legends. Maybe you could put a

larger version of this figure in the supplements. For example, split up the figure and present each glacier individually.

Thank you for this valuable feedback – we address this issue in two ways to ensure the concern is addressed. As suggested, the individual subplots for Figure 13 are now included in the Supplement as Figs. S13-S22. Additionally, we enhance the readability of each subplot by increasing font size, scale bars, and legend positioning. These changes are applied to the figure in the primary manuscript as well as to Figure 14.

The update Fig. 13 is provided in the response to Editor Comment #2, under Fig. R1.

Figure 14: This is a great overview of glacier changes for entire Greenland. In my opinion, the overall differences between the basins count more than the seasonal variability in this Figure. Therefore, I would recommend to use the same vertical axis scaling. This would highlight the differences in retreat for each basin at first glance (e.g. 6 km in SE versus 1.6 km in SW).

This suggestion is appreciated as well, and the vertical scales of each subfigure now share the same range, from -1 to 6.5 km. Additionally, the figure has been enlarged for better readability, as per the previous comment regarding Fig. 13, and Editor Comment #2.

The update Fig. 14 is provided in the response to Editor Comment #2, under Fig. R2.

Conclusion: Most readers tend to read the Abstract and Conclusion only. You could consider including the main findings of your study in the conclusion (e.g. accuracy 86.76± 1.43, number of studied glaciers, good performance compared to validation data, absolute retreat differences in all Greenlandic basins etc.).

This is an excellent suggestion, and additional highlights have been added to the conclusion.

The validation, data intercomparison results, and regional trend differences are now covered in sentences 3-4 of the conclusion: "The method is validated through a comprehensive data intercomparison with existing studies, and the results deviate by on average 86.76 ± 1.43 meters from the measured fronts. Regional trends show larger than average absolute retreat in SE Greenland, and new sub-seasonal trends are available for further investigation with the release of the 22,678 calving front lines generated across 66 Greenlandic glaciers."

**Anonymous Referee #2 Report #1**

**General Comments:**

The comments here concern the revised version of the manuscript by Cheng et al. on automated glacier calving front delineation in Greenland based on deep learning. The authors have made substantial revisions to the original manuscript in response to my and the other reviewer's comments. These revisions include clarifications, corrections, additional sections and references as well as updated and new figures. I believe this has significantly improved the earlier version of the manuscript. I especially welcome the expanded data intercomparison and analysis section regarding regional trends in glacier advance/retreat. I do think that some of these new analysis results merits a mention in the conclusion and abstract as well, this seems currently lacking. Some further minor modifications to the newly added text are also necessary (the line numbering of the track changes document is used) before publication.

We express our thanks to the reviewer for this feedback, and have integrated the contribution into the manuscript.

The abstract has been updated to include mentions of the data intercomparisons and regional analysis, and sentences 8-9 have been modified to read: "This improves on the state of the art in terms of the spatiotemporal coverage and accuracy of its outputs, and is validated through a comprehensive intercomparison with existing studies. The current implementation offers a new opportunity to explore sub-seasonal and regional trends on the extent of Greenland's margins, and supplies new constraints for simulations of the evolution of the mass balance of the Greenland Ice Sheet and its contributions to future sea level rise."

The conclusion now highlights the data intercomparison results and regional trend differences in more detail, covered by the addition of sentences 3-4: "The method is validated through a comprehensive data intercomparison with existing studies, and the results deviate by on average 86.76 ± 1.43 meters from the measured fronts. Regional trends show larger than average absolute retreat in SE Greenland, and new sub-seasonal trends are available for further investigation with the release of the 22,678 calving front lines generated across 66 Greenlandic glaciers."

**Specific Comments:**

Pg 1 Ln 1-4: It seems some logical steps are missing between these sentences. Something along the line of "documenting the evolution of calving front positions, for which satellite imagery forms the basis, is therefore important". The same seems to be the case before Pg 2 – Ln 3, where mention is made of calving front delineation without mentioning the main source data first: satellite imagery.

This contribution is much appreciated, and the suggestion has been integrated into both the abstract as well as the introduction.

The second line of the abstract now reads: "The documentation of these evolving calving front positions, for which satellite imagery forms the basis, is therefore important."

The sentence starting on Pg 2 Ln 3 now mentions satellite imagery as the basis of delineation as well: "While satellite imagery allows for the extensive documentation of this evolving constraint, most calving front delineation is still done with time-consuming manual labor"

Pg 2 Ln 10: "detecting calving front delineations": detecting the calving front

Done.

Pg 2 Ln 13: "by (Seale et al., 2010)": by Seale et al. (2010)

Fixed.

Pg 2 Ln 16: "for towards": remove towards

Fixed.

Pg 2 Ln 17: "...Kangerlussuaq.": Kangerlussuaq Glaciers.

Done.

Pg 2 Ln 19: "Jakobshavn": Jakobshavn Glacier

Done.

Pg 30 Ln 13: "to": too

"to" has been changed to "on" to enhance the clarity of this sentence clause.

[revised manuscript text omitted]